# A persistent homology-based algorithm for unsupervised anomaly detection in time series

**Alexandre Bois**                                    *alexandre.bois@ens-paris-saclay.fr*
*Université Paris Saclay, Université Paris Cité, ENS Paris Saclay, CNRS, SSA, INSERM, Centre Borelli*
*Gif-sur-Yvette, 91190, France*

**Brian Tervil**                                      *brian.tervil@ens-paris-saclay.fr*
*Université Paris Saclay, Université Paris Cité, ENS Paris Saclay, CNRS, SSA, INSERM, Centre Borelli*
*Gif-sur-Yvette, 91190, France*

**Laurent Oudre**                                     *laurent.oudre@ens-paris-saclay.fr*
*Université Paris Saclay, Université Paris Cité, ENS Paris Saclay, CNRS, SSA, INSERM, Centre Borelli*
*Gif-sur-Yvette, 91190, France*

**Reviewed on OpenReview:** *https://openreview.net/forum?id=H6ChfpM31L*

## Abstract

In this article, we propose a new algorithm for unsupervised anomaly detection in univariate time series, based on topological data analysis. It relies on delay embeddings and on the extraction of persistent cycles from the 1-dimensional persistent homology constructed from the distance to measure Rips filtration. This filtration makes it possible to identify 1-cycles (i.e. loops) corresponding to recurrent patterns by leveraging density information. Points in those cycles are considered as normal, and the algorithm can then assign an anomaly score to any point which is its distance to the normal set. In this paper, we describe the algorithm, make a theoretical study, and test it on several real-world and synthetic datasets, showing that it is competitive with state-of-the-art anomaly detection methods.

## 1 Introduction

Anomaly detection in time series is an important problem in data science, with applications in many fields such as healthcare (Ansari et al., 2017) and engineering (Woike et al., 2014). A time series is a sequence of real numbers $\mathbf{x} = (\mathbf{x}[i])_{1 \leq i \leq n}$ ($n$ will always denote the length of the time series). In the context of anomaly detection, $\mathbf{x}$ is assumed to be composed of a normal behavior and anomalies, i.e. points or sequences of points that differ from the normal behavior. More precisely, in several application contexts such as industrial monitoring or healthcare, the time series is usually assumed to be composed of some repetitive/frequent patterns, possibly of varying lengths (think for instance of an heartbeat in ECG data) among which some occur a large number of times (the *normal* ones) and some have significantly fewer occurrences (the *abnormal* ones) : see Figure 1 for an illustration.

Over the past years, several unsupervised anomaly detection algorithms have been developed from different research areas (Liu et al., 2008; Breunig et al., 2000; Goldstein & Dengel, 2012; Yeh et al., 2018; Boniol et al., 2021; Aggarwal & Aggarwal, 2017; Sakurada & Yairi, 2014; Malhotra et al., 2015; Li et al., 2007; Munir et al., 2018; Schölkopf et al., 1999) (see Paparrizos et al. (2022) and Schmidl et al. (2022) for a comprehensive review). Among them, some rely on a model and use the prediction or reconstruction error as an anomaly detector and some are based on clustering or machine learning techniques applied on the subsequences in order to detect outliers. For instance, LOF (Breunig et al., 2000) transforms the time series into a point cloud and studies the density of each point to assess whether or not they correspond to normal or abnormal behaviors. Some methods aim to find subsequences that represent the normal behavior and define anomalies as subsequences that differ from normality. The fact that the patterns can have different lengths, that there

can be multiple normal patterns and multiple occurrences of an anomaly, and noise make it difficult to build a universal anomaly detection algorithm (Paparrizos et al., 2022). The main differences between the approaches actually lie in the implied definition given to the notion of normality.

Topological data analysis (TDA), and more specifically persistent homology (Edelsbrunner & Harer, 2010; Boissonnat et al., 2018) is a set of techniques derived from algebraic topology, which allows to analyze the structure of data by constructing a sequence of simplicial complexes (a *filtration*). The *persistence diagram* sums up when connected components, loops or higher-dimensional simplices appear and disappear when going through the filtration. TDA has been applied to many fields (Chazal & Michel, 2021) including time series analysis. It is particularly adapted to study structured data such as time series with a periodic behavior (Perea & Harer, 2015; Emrani et al., 2014; Bonis et al., 2022; Bois et al., 2022), and methods from TDA benefit from stability theorems that guarantee a certain robustness to noise (Chazal et al., 2014; Anai et al., 2020). The most commonly used filtrations are the Čech and the Vietoris-Rips filtrations. As those filtrations are not robust to the presence of outliers, new variants based on the notion of distance to a measure were introduced in Chazal et al. (2011); Buchet et al. (2016); Anai et al. (2020), along with new stability theorems. Other relevant filtrations are defined in De Silva & Carlsson (2004); Bell et al. (2019). The authors of Anai et al. (2020); Ueda et al. (2022); Fernández et al. (2023) explain how sequences of persistence diagrams can be used for change detection or anomaly detection in time series. However, these methods seems more relevant for change detection than anomaly detection as it is not designed to handle typical cases such as punctual anomalies, multiple normal behaviors or repeated anomalies. Indeed, they consist in computing the persistence diagram corresponding to a sequence of growing subwindows of a time series and to compare consecutive diagrams to detect changes. Thus, when there are multiple patterns, changes would be detected at the first occurrence of each pattern but they would often not correspond to an anomaly. In the case of a repeated anomaly a change would only be detected at its first occurrence and the following ones would be missed. Moreover, this methods requires to compute many persistence diagrams which can lead to a high computation time.

In the case of periodic functions, the importance of 1-dimensional persistent homology (the study of loops in point clouds) was theoretically studied in Perea & Harer (2015) and it was applied to time series in Emrani et al. (2014); Perea et al. (2015) by transforming the data into a point cloud and considering the most important loop. By extension, 1-dimensional persistent homology is also relevant to study time series with repetitive patterns.

In a nutshell, our method consists in transforming the time series into a point cloud and extracting 1-cycles (i.e. loops) that are considered to correspond to normal patterns of the time series. We will use the Vietoris-Rips filtration associated to the empirical distance to a measure as described in Anai et al. (2020). Those cycles are identified on the persistence diagram because density information is used to construct the filtration. Once "normal cycles" have been extracted, an anomaly score is defined for each point of the embedding as its distance to the normal cycles. The specificity of our method is that it makes a global study of the delay embedding to find loops corresponding to whole patterns and also integrates local density information through the choice of the filtration, that is used to distinguish normal and abnormal points by reading the persistence diagram.

**Contributions.** In this paper, we apply methods from topological data analysis to unsupervised anomaly detection in time series. Our contributions are listed below.

- We propose a model of time series that makes it possible to formally define the anomaly detection problem. It includes the possibility of having multiple normal or abnormal behaviors, repeating anomalies and noise. We show examples of real-life time series that fit this model.

- We present a new method for anomaly detection based on persistent homology. The method uses a delay embedding and the Vietoris-Rips filtration associated to the empirical distance to a measure to find 1-cycles made of dense points and identify them to the normal behavior.

- We use a property of this filtration to deal with a subset of the point cloud while keeping information from the whole data in order to significantly decrease computation time.

- We use our model to derive an upper bound on the interleaving distance between the filtration used in our algorithm (with discrete, noisy data with anomalies) and a filtration obtained in an ideal situation with a continuous signal without noise or anomalies.

- We study the behavior of our algorithm with different parameters and the influence of noise, and show that it is competitive to state-of-the-art anomaly detection methods on different real-world datasets.

The article is organized as follows. in Section 2, we introduce our model of time series and our definition the anomaly detection problem, along with the theoretical background required for the rest of the paper. Section 3 describes our algorithm. Section 4 contains the theoretical study. In Section 5, we present the 13 state-of-the-art anomaly detection methods and real-world and synthetic datasets. In Section 6, we use these datasets to study the behavior of our algorithm with different parameters and the influence of noise, compare it to state-of-the-art anomaly detection methods. Note that section 2.2, 2.3 and 4.1 present already existing objects and results. Every other sections describe new research.

## 2 Background and problem formulation

In this section, we give the mathematical background required to understand the studied problem and the proposed algorithm. Notations that are used throughout the paper are listed in Table 1.

### 2.1 Model and problem

In this subsection, we introduce a model for time series which makes it possible to formally define the anomaly detection problem.

Let $\mathbf{x} = (\mathbf{x}[t])_{t \in [0,1]}$ and $\boldsymbol{\varepsilon} = (\boldsymbol{\varepsilon}[t])_{t \in [0,1]}$ be real-valued functions defined on $[0,1]$. We call $\mathbf{x}$ the *signal* and $\boldsymbol{\varepsilon}$ is called the *noise*.

**Remark 1.** *We will always use the term signal for continuous real-valued functions, and time series for finite sequences of real numbers.*

We use a convolutional sparse coding model (Papyan et al., 2017) with an additional hypothesis to describe $\mathbf{x}$ as a sequence of *atoms*, which are either normal or abnormal. Formally, we assume that $\mathbf{x}$ can be written in the following way:

$$\mathbf{x} = \sum_{i=1}^{M_{\mathbf{n}}} \mathbf{n}_i * \boldsymbol{\eta}_i + \sum_{i=1}^{M_{\mathbf{a}}} \mathbf{a}_i * \boldsymbol{\alpha}_i. \tag{1}$$

The $\mathbf{n}_i$ and $\mathbf{a}_i$ are respectively normal and abnormal atoms of length $l_{\mathbf{n}_i} < 1$ (resp. $l_{\mathbf{a}_i}$), i.e. continuous functions on $[0,1]$ with support $[0, l_{\mathbf{n}_i}]$ (resp. $[0, l_{\mathbf{a}_i}]$) and such that $\mathbf{n}_i(0) = \mathbf{n}_i(l_{\mathbf{n}_i}) = 0$ (resp. $\mathbf{a}_i(0) = \mathbf{a}_i(l_{\mathbf{a}_i}) = 0$). There are $M_{\mathbf{n}}$ normal atoms and $M_{\mathbf{a}}$ abnormal atoms. The $\boldsymbol{\eta}_i$ and $\boldsymbol{\alpha}_i$ are the *activations*: binary functions on $[0,1]$ that take the value 1 only a finite number of times. Thus, $\mathbf{n}_i * \boldsymbol{\eta}_i$ is a signal where an *occurrence* of $\mathbf{n}_i$ starts at each time $t$ where $\boldsymbol{\eta}_i[t] = 1$. For each $i$, the number of occurrences of atom $\mathbf{n}_i$ (resp. $\mathbf{a}_i$) is denoted by $k_{\mathbf{n}_i} = ||\boldsymbol{\eta}_i||_0$ (resp. $k_{\mathbf{a}_i} = ||\boldsymbol{\alpha}_i||_0$), where $||.||_0$ is the number of non-zero values of a function or vector.

Finally, we define the normal and abnormal parts of the signal as

$$\mathbf{x_n} = \sum_{i=1}^{M_{\mathbf{n}}} \mathbf{n}_i * \boldsymbol{\eta}_i \text{ and } \mathbf{x_a} = \sum_{i=1}^{M_{\mathbf{a}}} \mathbf{a}_i * \boldsymbol{\alpha}_i$$

Table 1: Notations

| Notation | Description |
|---|---|
| $t$ | Time |
| $n$ | Length of the time series. |
| $d$ | Dimension of the time delay embedding. |
| $\tau$ | Delay of the time delay embedding. |
| $\mathbf{x} = (\mathbf{x}[t])_{t \in [0,1]}$ | Signal: a continuous function defined on $[0,1]$. |
| $\hat{\mathbf{x}} = (\hat{\mathbf{x}}[i])_{1 \le i \le n}$ | Time series: a finite uniform subsampling of $\mathbf{x}$. |
| $\mathbf{n}_i$ | Normal atom $i$. |
| $\mathbf{a}_i$ | Abnormal atom $i$. |
| $\mathbf{x_n}$ (resp. $\mathbf{x_a}$) | Normal (resp. abnormal) part of the signal. |
| $\hat{\mathbf{x}}_\mathbf{n}$ (resp. $\hat{\mathbf{x}}_\mathbf{a}$) | Normal (resp. abnormal) part of the time series. |
| $M_{\mathbf{n}_i}$ (resp. $M_{\mathbf{a}_i}$) | Number of normal (resp abnormal) atoms. |
| $\boldsymbol{\eta}_i$ (resp. $\boldsymbol{\alpha}_i$) | Activations of $\mathbf{n}_i$ (resp. $\mathbf{a}_i$). |
| $k_{\mathbf{n}_i}$ (resp. $k_{\mathbf{a}_i}$) | Number of occurrences of $\mathbf{n}_i$ (resp. $\mathbf{a}_i$). |
| $l_{\mathbf{n}_i}$ (resp. $l_{\mathbf{a}_i}$) | Length of atom $\mathbf{n}_i$ (resp. $\mathbf{a}_i$). |
| $k_\mathbf{n}$ | Number of normal occurrences: $\sum_{i=1}^{M_{\mathbf{n}_i}} k_{\mathbf{n}_i}$. |
| $k_\mathbf{a}$ | Number of abnormal occurrences: $\sum_{i=1}^{M_{\mathbf{a}_i}} k_{\mathbf{a}_i}$. |
| $k_\mathbf{x}$ | Total number of occurrences: $k_\mathbf{n} + k_\mathbf{a}$. |
| $V = (V_\alpha)_{\alpha \in \mathbb{R}^+}$ | A filtration. |
| $\mathbb{V}$ | Persistent homology module corresponding to the filtration $V$. |
| $\mathsf{Diag}(\mathbb{V})$ | Persistence diagram corresponding to the persistence module $\mathbb{V}$. |
| $X_{d,\tau}$ or $X$ | Time delay embedding of a signal or time series $\mathbf{x}$, with dimension $d$ and delay $\tau$ (or, in Section 2.2, a subset of E). |
| $\mu_B$ (where $B \subset E$) | A probability measure on $E$ with support included in $B$. |
| $d_{\mu_B, m}$ | Distance function to the measure $\mu_B$ with parameter $m$. |
| $d_i$ | Interleaving pseudo-distance. |
| $d_b$ | Bottleneck distance. |
| $W_2$ | Wasserstein distance with quadratic cost. |
| $d_H$ | Hausdorff distance. |
| $p \in \mathbb{N}$ | Parameter of the weighted filtrations. |
| $m = \frac{q}{\mathsf{Card}(\hat{X})} \in ]0,1[$ (with $q \in \mathbb{N}$) | Parameter $m$ of the DTM. |
| $\mathsf{Cech}[X, f, p]$ (where $B \subset E$) | Weighted Čech filtration with parameters $(X, f, p)$. with function $f$, and parameters $m, p$. |
| $\mathsf{Rips}[X, f, p]$ (where $B \subset E$) | Weighted Rips filtration with parameters $(X, f, p)$. with function $f$, and parameters $m, p$. |

along with the numbers $k_\mathbf{n} = \sum_{i=1}^{M_{\mathbf{n}_i}} k_{\mathbf{n}_i}$, $k_\mathbf{a} = \sum_{i=1}^{M_{\mathbf{a}_i}} k_{\mathbf{a}_i}$, and $k_\mathbf{x} = k_\mathbf{n} + k_\mathbf{a}$.

Let $n > 1 \in \mathbb{N}$. In the rest of the paper, we will always denote by $\hat{\mathbf{s}}$ the uniform subsampling of size $n$ of any given signal $\mathbf{s} = (\mathbf{s}[t])_{t \in [0,1]}$, that is $\hat{\mathbf{s}} = (\hat{\mathbf{s}}[j])_{1 \le j \le n}$, where $\hat{\mathbf{s}}[j] = \mathbf{s}[\frac{j-1}{n-1}]$. All the above signals, atoms, activations and quantities can be defined the same way in the discrete case starting from

$$\hat{\mathbf{x}} = \sum_{i=1}^{M_\mathbf{n}} \hat{\mathbf{n}}_i * \hat{\boldsymbol{\eta}}_i + \sum_{i=1}^{M_\mathbf{a}} \hat{\mathbf{a}}_i * \hat{\boldsymbol{\alpha}}_i.$$

With this model, we can formally define the anomaly detection problem.

**Definition 1** (Anomaly detection problem). *The anomaly detection problem consists in finding the set of integers $i \in [1, n]$ such that $\hat{\mathbf{x}}_\mathbf{a}[i] \neq 0$, which means finding all the discrete abnormal activations and atom*

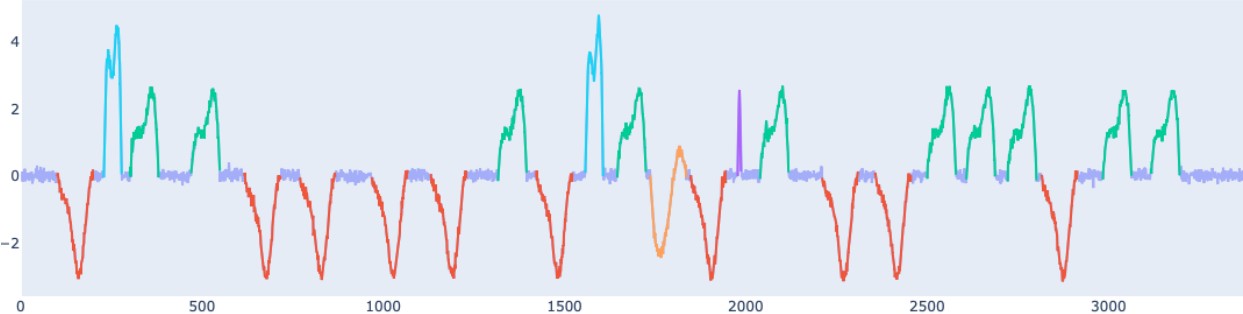

Figure 1: An example time series, with normal patterns in green and red (and the null parts in dark blue) and three abnormal ones (in blue, orange and purple), with Gaussian noise of amplitude 0.1.

*lengths.*

We now make some assumptions related to the model in the context of anomaly detection. The first assumption states that all the $\mathbf{n}_i * \boldsymbol{\eta}_i$ and $\mathbf{a}_i * \boldsymbol{\alpha}_i$ have disjoint supports, which implies that each point $\mathbf{x}[t]$ is either normal or abnormal.

**Assumption 1.** *if we denote by $(\boldsymbol{\zeta}_i)_{1 \leq i \leq M_\mathbf{n} + M_\mathbf{a}}$ the list of all the activations $\boldsymbol{\eta}_i$ and $\boldsymbol{\alpha}_i$, the assumption is:*

$$\forall i, \forall t \in [0, 1], \quad \boldsymbol{\zeta}_i[t] \neq 0 \Rightarrow \forall j \neq i, \forall s \in [t, t + l_{\boldsymbol{\zeta}_i}], \quad \boldsymbol{\zeta}_j[s] = 0.$$

**Assumption 2.** *The observed data $\hat{\mathbf{y}}$ (the time series) is a uniform subsampling of a noisy version of the signal $\mathbf{x}$: $\hat{\mathbf{y}} = \widehat{\mathbf{x} + \boldsymbol{\varepsilon}}$.*

**Assumption 3.** *For the anomaly detection problem, we will assume that $\max_i(k_{\mathbf{a}_i})$ is significantly lower than $\min_i(k_{\mathbf{n}_i})$, which means that anomalies are rare atoms.*

**Remark 2.**
- *Assumption 1 implies that all the atoms start and end at zero. This will be crucial for our method as we will transform the signal into a curve on which we will look for loops. The assumption ensures that each pattern corresponds to at least one loop.*

- *Assumption 2 corresponds to the situation where a quantity assumed to vary continuously is measured at regular intervals, with noise corresponding to errors or perturbations. Note that the only assumption we will make on the noise is that $\|\boldsymbol{\varepsilon}\|_\infty$ is small compared to the variations of the signal. Thus, it could represent small variations at each occurrence of an atom, of noise due to the data acquisition process. Moreover, introducing $\boldsymbol{\varepsilon}$ also makes it possible to study the problem without the disjoint supports assumption as long as the overlap between supports causes a small enough change in the infinity norm of $\mathbf{x}$.*

- *Assumption 3 corresponds to the specificity of the anomaly detection problem (compared, for example, to the pattern detection problem).*

Our model is illustrated on Figure 1. All the atoms start and end at zero and have disjoint supports. The green and red ones each occur ten times: they are the normal atoms. The blue one occurs twice, the orange and purple one each occur once. Note that atoms can have significantly different lengths.

Figure 2 show two example of time series that fit our model, from real-world datasets included in the TSB-UAD suite. The first one comes from the ECG dataset, which is a standard electrocardiogram dataset with anomalies that represent ventricular premature contractions. This shows that our model is relevant to study structured activities that present repetitive behaviors over time where anomalies in the real-world manifest as anomalies on the signal (here: pathologies cause premature ventricular contractions that appear as anomalies on the ECG). The second one comes from the NAB dataset, which is composed of labeled

real-world and artificial time series including AWS server metrics, online advertisement clicking rates, real time traffic data, and a collection of Twitter mentions of large publicly-traded companies.

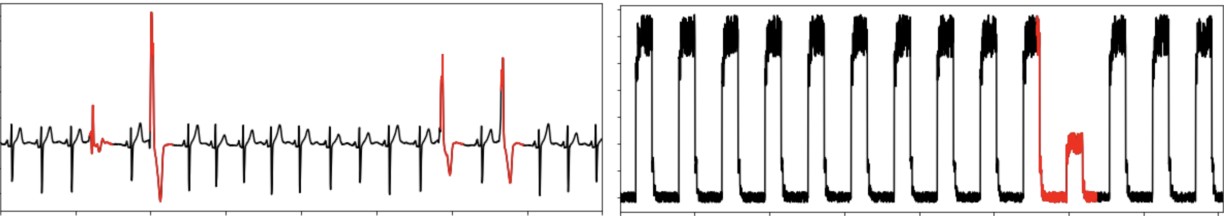

Figure 2: Two example of real-world time series that fit our model, with anomalies in red. **Left:** Time series from the ECG dataset. **Right:** Time series from the NAB dataset. See Paparrizos et al. (2022) for details on the datasets.

## 2.2 Topological data analysis background

In this subsection, we introduce the objects and results from TDA (mostly from Anai et al. (2020)) that will be used in the rest of the paper. See Boissonnat et al. (2018); Edelsbrunner & Harer (2010); Chazal & Oudot (2008); Chazal et al. (2012; 2014); Anai et al. (2020) for more complete background.

### 2.2.1 Filtrations and persistence modules

For the rest of this paper, let $\mathbb{R}^d$ be endowed with the Euclidean norm $||\cdot||$.

**Definition 2** (Simplicial complexes). *A $k$-simplex on a set $X \subset \mathbb{R}^d$ is an unordered tuple $\sigma = [x_0, ..., x_k]$ of $k+1$ distinct elements of $X$. The elements $x_0, ..., x_k$ are called the vertices of $\sigma$. If each vertex of a simplex $\rho$ is also a node of $\sigma$, then $\rho$ is called a face of $\sigma$. A simplicial complex $K$ is a set of simplices such that any face of a simplex of $K$ is a simplex of $K$.*

**Definition 3** (Filtrations and interleavings). *A filtration is a family $V = (V_\alpha)_{\alpha \in \mathbb{R}^+}$ of topological spaces or simplicial complexes such that $\alpha \leq \beta \Rightarrow V_\alpha \subset V_\beta$. For $\epsilon \geq 0$, two filtrations $V$ and $W$ are called $\epsilon$-interleaved if for all $\alpha \geq 0$, $V_\alpha \subset W_{\alpha+\epsilon}$ and $W_\alpha \subset V_{\alpha+\epsilon}$. The interleaving pseudo-distance is then defined as*

$$d_i(V, W) = \inf\{\epsilon \geq 0 | V \text{ and } W \text{ are } \epsilon\text{-interleaved}\}.$$

In our algorithm, the sets $V_\alpha$ will be *simplicial complexes* in $\mathbb{R}^d$.

In the case of a filtration of simplicial complexes over a finite set, only a finite number of values of $\alpha$ correspond to a strict inclusion in the filtration. We call *filtration value* of a simplex $\sigma$ the lowest $\alpha$ such that $\sigma \in V_\alpha$.

**Definition 4** (Persistence modules). *Let $K$ be a field. A persistence module is a family $\mathbb{V} = (\mathbb{V}_b, (v_a^b)_{0 \leq a \leq b})$ where the $\mathbb{V}_b$ are $K$-vector spaces and the $v_a^b$ are linear maps $\mathbb{V}_a \to \mathbb{V}_b$ such that for all real numbers $a \leq b \leq c$, $v_a^a = Id$ and $v_a^b \circ v_b^c = v_a^c$.*

**Definition 5** (Morphisms and interleavings of persistence modules). *Let $\epsilon \geq 0$. An $\epsilon$-morphism between two persistence modules $\mathbb{V}$ and $\mathbb{W}$ is a family of linear maps $\phi = (\phi_\alpha : \mathbb{V}_\alpha \to \mathbb{W}_{\alpha+\epsilon})_{\alpha \in \mathbb{R}^+}$ such that the following diagram commutes for all $a \leq b$ :*

$$
\begin{array}{ccc}
\mathbb{V}_a & \xrightarrow{v_a^b} & \mathbb{V}_b \\
\downarrow{\phi_a} & & \downarrow{\phi_b} \\
\mathbb{W}_{a+\epsilon} & \xrightarrow{v_{a+\epsilon}^{b+\epsilon}} & \mathbb{W}_{b+\epsilon}
\end{array}
$$

*If $\epsilon = 0$ and if all the $\phi_\alpha$ are isomorphisms, $\phi$ is called an isomorphism of persistence modules.*

*An $\epsilon$-interleaving between two persistence modules $\mathbb{V}$ and $\mathbb{W}$ is a pair $(\phi, \psi)$ of morphisms such that the following diagrams commute for all $a \in \mathbb{R}$:*

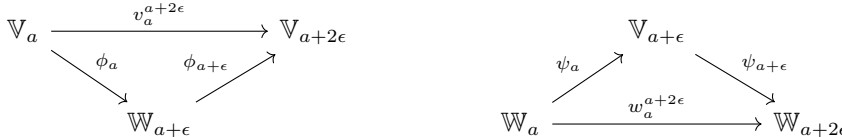

*The interleaving pseudo-distance is then defined as*

$$d_i(\mathbb{V}, \mathbb{W}) = \inf\{\epsilon \geq 0 | \mathbb{V} \, and \, \mathbb{W} \, are \, \epsilon\text{-}interleaved\}.$$

### 2.2.2 Persistent homology

Applying the (singular or simplicial) homology functor to a filtration and its inclusion maps gives a persistence module of homology groups $\mathbb{V} = (H_i(V_\alpha))_{\alpha \in \mathbb{R}^+}$ for each dimension $i \in [0, d]$. If for all $a \leq b$, $v_a^b$ is of finite rank, then $\mathbb{V}$ is called *q-tame* and $\mathbb{V}$ can be decomposed into a direct sum of *interval modules* $\bigoplus_j \mathbb{I}_{[\alpha_{l(j)}, \alpha_j)}$, where the $[\alpha_{l(j)}, \alpha_j)$ are left closed, right open intervals and each $\mathbb{I}_{[\alpha_{l(j)}, \alpha_j)}$ is a basic persistence module that represent a feature that appears at index $\alpha_{l(j)}$ and disappears at index $\alpha_j$ (see Chazal et al. (2012) for more details). In this case, the *persistence diagram* of $\mathbb{V}$ can be defined. The persistence diagram $\mathsf{Diag}(\mathbb{V})$ of a $i$-dimensional persistent homology module $\mathbb{V}$ is a multiset $\{(\alpha_{l(j)}, \alpha_j)\}$, such that a new homology class that appears in $H_i(V_{\alpha_{l(j)}})$ disappears in $H_i(V_{\alpha_j})$ ($\alpha_{l(j)}$ is called its birth date and $\alpha_j$ its death date, if the component never dies, $\alpha_j = \infty$). We call $\alpha_j - \alpha_{l(j)}$ the *persistence* of the point. By convention and to be able to define the *bottleneck distance*, we include every points $(\alpha, \alpha)$ to the diagram, with infinite multiplicity. 0-dimensional homology classes are connected components, and 1-dimensional classes are loops.

**Definition 6.** *The bottleneck distance between two persistence diagrams $D$ and $D'$ is defined as:*

$$d_b(D, D') = \inf_{\gamma \in \Gamma(D, D')} \sup_{(\alpha, \beta) \in D} ||(\alpha, \beta) - \gamma((\alpha, \beta))||_\infty$$

*where $\Gamma(D, D')$ is the set of bijections from $D$ to $D'$, and where $||(\alpha, \beta) - (\alpha', \beta')||_\infty = |\alpha - \alpha'|$ when $\beta = \beta' = +\infty$.*

Note that if two filtrations are $\epsilon$-interleaved, then the corresponding persistence modules are also $\epsilon$-interleaved. Moreover, the *isometry theorem* (Chazal et al., 2012) states that $d_i(\mathbb{V}, \mathbb{W}) = d_b(\mathsf{Diag}(\mathbb{V}), \mathsf{Diag}(\mathbb{W}))$ for q-tame modules $\mathbb{V}$ and $\mathbb{W}$. We will implicitly use this in Section 4, as a bound on the interleaving distance between filtrations induces a bound on the bottleneck distance between persistence diagrams, which are the objects studied by our algorithm.

### 2.2.3 DTM filtrations

Let $X \subset \mathbb{R}^d$, $f : X \to \mathbb{R}^+$ and $p \geq 1$ be a real number. For $x \in X$ and $\alpha \in \mathbb{R}^+$, we define $r_x(\alpha)$ as:

$$r_x(\alpha) = \begin{cases} -\infty & \text{if } \alpha < f(x) \\ (\alpha^p - f(x)^p)^{\frac{1}{p}} & \text{otherwise} \end{cases}$$

or $r_x(\alpha) = \alpha$ for $p = \infty$. Now, let us denote by $\bar{B}_f(x, \alpha)$ the closed ball $\bar{B}(x, r_x(\alpha))$ (by convention, a ball is empty if its radius is $-\infty$). We can now define the *weighted Čech filtration* as a union of growing balls.

**Definition 7.** *With the above notations, the weighted Čech filtration with parameters $(X, f, p)$, $\mathsf{Cech}[X, f, p]$, is defined by:*

$$\mathsf{Cech}[X, f, p]_\alpha = \bigcup_{x \in X} \bar{B}_f(x, \alpha).$$

Let $\mathcal{N}(\mathsf{Cech}[X, f, p]) = (\mathcal{N}(\mathsf{Cech}[X, f, p]_\alpha))_{\alpha \geq 0}$ denote the nerve of the above cover of $X$ by closed Euclidean balls, i.e. the filtration of simplicial complexes whose $(i-1)$-simplices (with $i - 1 \leq d$) are defined by

$$[x_1, \ldots, x_i] \in \mathcal{N}(\mathsf{Cech}[X, f, p]_\alpha) \iff \bigcap_{j=1}^{i} \bar{B}_f(x_j, \alpha) \neq \emptyset.$$

The persistent nerve theorem (Chazal & Oudot (2008), Lemma 3.4) states that the persistence (singular) homology module associated to the weighted Čech filtration is isomorphic to the persistence (simplicial) homology module associated to its nerve. The latter is computable in practice (Edelsbrunner & Harer, 2010; Boissonnat et al., 2018).

**Definition 8.** *With the above notations, the weighted Vietoris-Rips filtration with parameters $(X, f, p)$, $\mathsf{Rips}[X, f, p]$ is the flag complex of $\mathcal{N}(\mathsf{Cech}[X, f, p])$, i.e. the filtered simplicial complex such that each vertex (or 0-simplex) $x \in X$ has filtration value $f(x)$, and such that for $2 \leq i \leq d + 1$:*

$$[x_1, \ldots, x_i] \in \mathsf{Rips}[X, f, p]_\alpha \iff \forall (j, k), \ \bar{B}_f(x_j, \alpha) \cap \bar{B}_f(x_k, \alpha) \neq \emptyset.$$

The Vietoris-Rips complex is easier to compute than the Čech complex in practice because one only needs to know the filtration values of the 0 and 1 simplices to characterize the whole filtration. The following proposition gives this value in the case $p = 1$, which we will use in this paper (see Anai et al. (2020) for values when $p = 2$ or $p = \infty$).

**Proposition 1** (Anai et al. (2020)). *Let $x, y \in X$. The filtration value of $[x]$ in $\mathsf{Rips}[X, f, 1]$ is $f(x)$, and the filtration value of $[x, y]$ in $\mathsf{Rips}[X, f, 1]$ is:*

$$\begin{cases} \mathsf{max}(f(x), f(y)) & \text{if } ||x - y|| < |f(x) - f(y)| \\ \frac{||x - y|| + f(x) + f(y)}{2} & \text{otherwise.} \end{cases}$$

In the rest of this paper, the function $f$ will always be a *distance to measure function*. Those functions were introduced in Anai et al. (2020) to make weighted filtrations robust to outliers.

**Definition 9** (DTM). *Let $\mu$ be a probability measure on $\mathbb{R}^d$ and let $m \in [0, 1[$. Let $\delta_{\mu, m}$ be the function defined on $\mathbb{R}^d$ by $\delta_{\mu, m}(x) = \mathsf{inf}\left\{r > 0 \mid \mu(\bar{B}(x, r)) > m\right\}$. The distance to measure (DTM) $\mu$ with parameter $m$ is the function defined on $\mathbb{R}^d$ by:*

$$d_{\mu, m}(x) = \sqrt{\frac{1}{m} \int_0^m \delta_{\mu, u}^2(x) du}.$$

If $\hat{X}$ is a finite subset of $\mathbb{R}^d$, we denote by $\mu_{\hat{X}}$ the empirical measure on $\hat{X}$. If $m = \frac{q}{\mathsf{Card}(\hat{X})}$ with $q \in \mathbb{N}$, then:

$$\forall x \in \mathbb{R}^d, \ d_{\mu_{\hat{X}}, m}(x) = \sqrt{\frac{1}{q} \sum_{i=1}^{q} ||x - NN^{(i)}(x)||^2}$$

where $NN^{(i)}(x)$ is the $i^{th}$ nearest neighbor to $x$. The *DTM Čech and Vietoris-Rips filtrations* for a measure $\mu$ and a parameter $m$ are defined as the filtrations $\mathsf{Cech}[X, d_{\mu, m}, p]$ and $\mathsf{Rips}[X, d_{\mu, m}, p]$.

### 2.3 Delay embeddings

A *delay embedding* is a way of transforming a signal/time series into a curve/point cloud of chosen dimension $d$. Delay embeddings come from the field of dynamical systems, with strong theoretical guaranties (Takens, 1981; Sauer et al., 1991).

**Definition 10.** *Using the above notations, we define the delay embedding of a signal* $\mathbf{s}$ *with dimension $d \geq 2$ and delay $\tau \in \mathbb{R}$ as the following curve in $\mathbb{R}^d$:*

$$S_{d,\tau} = (S_{d,\tau}[t])_{t \in [0,1-(d-1)\tau]} = ((\mathbf{s}[t], \mathbf{s}[t+\tau], \ldots, \mathbf{s}[t+(d-1)\tau]))_{t \in [0,1-(d-1)\tau]}.$$

*The delay embedding of the time series $\hat{\mathbf{s}}$ with dimension $d \geq 2$ and delay $\hat{\tau} \in \mathbb{N}$ is the following point cloud in $\mathbb{R}^d$:*

$$\begin{aligned}
\hat{S}_{d,\hat{\tau}} &= (\hat{S}_{d,\hat{\tau}}[i])_{1 \leq i \leq n-(d-1)\hat{\tau}} \\
&= ((\hat{\mathbf{s}}[i], \hat{\mathbf{s}}[i+\hat{\tau}], \ldots, \hat{\mathbf{s}}[i+(d-1)\hat{\tau}]))_{1 \leq i \leq n-(d-1)\hat{\tau}} \\
&= \left(\left(\mathbf{s}\left[\frac{i}{n-1}\right], \mathbf{s}\left[\frac{i+\hat{\tau}}{n-1}\right], \ldots, \mathbf{s}\left[\frac{i}{n-1} + (d-1)\frac{\hat{\tau}}{n-1}\right]\right)\right)_{0 \leq i \leq n-1-(d-1)\hat{\tau}}.
\end{aligned}$$

Note that if $\tau = \frac{\hat{\tau}}{n-1}$ then $\hat{S}_{d,\hat{\tau}} \subset S_{d,\tau}$. In the rest of the paper, we fix $d$ and $\hat{\tau}$ for time series and use $d$ and $\tau = \frac{\hat{\tau}}{n-1}$ for signals. When there is no ambiguity, we use $S$ and $\hat{S}$ instead of $S_{d,\tau}$ and $\hat{S}_{d,\hat{\tau}}$.

In the following sections, we will use the embeddings $X, X_{\mathbf{n}}, X_{\mathbf{a}}$ of the signals $\mathbf{x}, \mathbf{x_n}, \mathbf{x_a}$, the embeddings $\hat{X}, \hat{X}_{\mathbf{n}}, \hat{X}_{\mathbf{a}}$ of the time series $\hat{\mathbf{x}}, \hat{\mathbf{x}}_{\mathbf{n}}, \hat{\mathbf{x}}_{\mathbf{a}}$, and their noisy versions ($Y = X + \mathcal{E}$ being the notation for the embedding of the noisy signal $\mathbf{y} = \mathbf{x} + \boldsymbol{\varepsilon}$). Figure 3 shows a delay embedding of the time series from Figure 1. Each colored loop on Figure 3 corresponds to the atom of the same color on Figure 1. The presence of noise makes loops corresponding to normal atoms (in green and red) thicker, as these atoms have more occurrences.

## 3 Method

In this section, we describe our algorithm for unsupervised anomaly detection. We start by explaining the main ideas behind the algorithm, and then describe its four steps: compute the DTM Rips filtration of a subset of the delay embedding, identify the normal 1-cycles, extract them, and compute the anomaly scores. We use the above notations for all parameters and mathematical objects.

### 3.1 Motivation

The solution to the anomaly detection problem described in Section 2 is a binary vector of length $n$: each point $\hat{\mathbf{y}}[i]$ of the time series is either normal or abnormal. Our algorithm, like most anomaly detection algorithms (Paparrizos et al., 2022; Schmidl et al., 2022), works by giving an anomaly score to each $\hat{\mathbf{y}}[i]$ which should be high if the point is abnormal. A binary answer can then be obtained by choosing a threshold over which points are considered abnormal. We do not give a specific method to choose the threshold, as it depends on the application.

Our algorithm works by studying a delay embedding $\hat{Y}$ of $\hat{\mathbf{y}}$ to identify a set of points as normal. Then, each point of $\hat{Y}$ is given an anomaly score which is its distance to the set of normal points. Finally, we get an anomaly score for $\hat{\mathbf{y}}[i]$ by averaging the scores of all points in $\hat{Y}$ of which $\hat{\mathbf{y}}[i]$ is a coordinate.

The idea behind our algorithm is that, if we consider our model of signals from Section 2 and if $(d-1)\tau$ is small enough, Assumption 1 implies that the embedding of each occurrence of atom should starts and ends

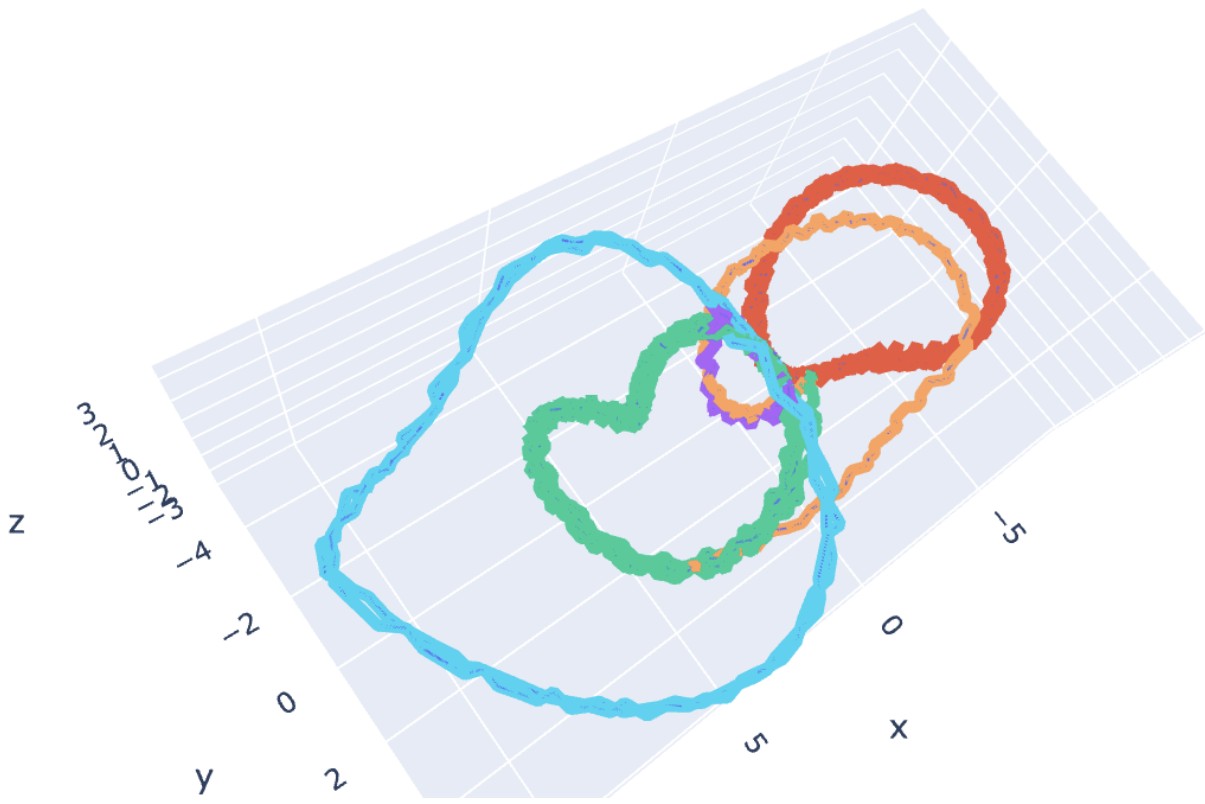

Figure 3: Delay embedding of the time series from Figure 1 (PCA in 3D), with $d = 10$ and $\hat{\tau} = 4$. Colors corrrespond to the colors of atoms on Figure 1.

at $(0, \ldots, 0)$, so it has at least one loop. 1-dimensional persistent homology can be used to detect those loops in $\hat{Y}$ as homology classes of 1-cycles. Each point from a normal atom $\mathbf{n_i}$ will have $k_{\mathbf{n_i}}$ occurrences and each point from an abnormal atom $\mathbf{a_i}$ will have $k_{\mathbf{a_i}}$ occurrences, so if $\mathsf{max}(k_{\mathbf{a_i}}) < q \leq \mathsf{min}(k_{\mathbf{n_i}})$ (see Assumption 3), then $d_{\mu_{\hat{Y}}, \frac{q}{n-(d-1)\hat{\tau}}}$ should be close to zero for normal points and strictly higher for abnormal points, which would make it possible to discriminate the corresponding 1-cycles on the persistence diagram, as their birth date will be different. Our set of normal points is then the set of 1-cycles detected as normal.

**Remark 3.** *One could argue that using 1-cycles is useless because points could be studied individually using their filtration value (i.e. the birth dates on the 0-dimensional persistence diagram) which is the opposite of a measure of the density of $\hat{Y}$ around each point. This approach would then be similar to the LOF algorithm (Breunig et al., 2000). However, in practice, data do not fit our ideal model (there can be noise, differences between occurrences of the same atoms or non-disjoint supports) and there is no way to be certain that $\mathsf{max}(k_{\mathbf{a_i}}) < q \leq \mathsf{min}(k_{\mathbf{n_i}})$ so the filtration values can take a range of values and choosing a threshold would be hard. Moreover, an abnormal atom with slow variations will give dense points in $\hat{Y}$ which would make the 0D approach or LOF fail. Considering 1D persistent homology and considering only cycles with a high persistence makes the choice of normal points easier by focusing on a few cycles (and thus a few birth dates) corresponding to structured components of $\hat{Y}$. It also makes it possible to eliminate atoms with slow variations (because the corresponding cycles have low persistence).*

## 3.2 Algorithm description

The input of our algorithm is a time series $\mathbf{y}$. There are five parameters that will be described below: $d, \hat{\tau}, n_{points}, q, n_{diag}$.

### 3.2.1 Delay embedding, subsampling and DTM Rips filtration

We start by choosing a dimension $d$ and a delay $\hat{\tau}$ and computing the delay embedding $\hat{Y} = \hat{Y}_{d,\hat{\tau}}$. As point clouds can be very large (they have $n - (d-1)\hat{\tau}$ points), computing persistent homology can be too long for the algorithm to be used in practice ($O(n^3)$ in the worst case for 1D persistent homology and cycle extraction (Boissonnat et al., 2018; Edelsbrunner et al., 2002)). To solve this problem, we choose a number $n_{points}$ and take a subsample $\tilde{Y}$ of $\hat{Y}$ made of $n_{points}$ points. In practice, we use a greedy method: we start with a random point and, until we have $n_{points}$ points, add the furthest one to the set of already chosen points.

We then choose an integer $q > 1$ and set the DTM parameter $m$ to $m = \frac{q}{\mathsf{Card}(\tilde{Y})} \in ]0,1[$ and compute the filtration $V = \mathsf{Rips}[\tilde{Y}, d_{\mu_{\hat{Y}},m}, 1]$ with $p = 1$. We chose to always use $p = 1$ because among the three possible values for which Anai et al. (2020) provides explicit formulas for the filtration values (1, 2, and $\infty$), $p = 1$ gives the strongest theoretical guarantees (see section 4) and the highest value to edges, which can give more persistent cycles that are easier to identify on the diagram.

**Remark 4.** *Note that the filtration values of points in $\tilde{Y}$ are their original values from $d_{\mu_{\hat{Y}_{d,\hat{\tau}}},m}$ (subsampling does not change the fact that $V$ is a filtration). This is important to keep the density information from the whole point cloud when subsampling (otherwise, the effect of the number of occurrences of normal atoms would disappear).*

### 3.2.2 Identifying normal cycles

The second step consists in identifying normal 1-cycles by reading the persistence diagram, and extracting those cycles. Let $\mathsf{Diag}(\mathbb{V})$ be the persistence diagram corresponding to the filtration $V$. To identify normal cycles, we propose an algorithm that relies on the choice of two thresholds: the persistence threshold (we focus on the most persistent points, which describe important structures), and the birth date threshold (among those points, we consider those with a birth date above the cycle to be abnormal).

To choose the persistence threshold, we sort the persistence of all points by decreasing order in a list $L$, find the index $i$ such that $L[i] - L[i+1]$ is maximal, and keep points corresponding to indices from 1 to $i + n_{diag}$, where $n_{diag}$ is a chosen parameter. See Algorithm 1 for a formal description. We will use $n_{diag} \geq 2$ in practice, to keep at least three points and thus to be able to compare at least two differences in birth dates.

---

**Algorithm 1** PersistenceThreshold (lists start at index 1)

**Require:** $\mathsf{Diag}(\mathbb{V})$, $n_{diag}$
1: $L \leftarrow$ [d-b for (b,d) in $\mathsf{Diag}(\mathbb{V})$]
2: Sort $L$ in decreasing order.
3: $N \leftarrow \mathsf{length}(L)$
4: $\mathsf{Diffs} \leftarrow [L[i] - L[i+1]$ for $i = 1 \ldots N-1]$
5: $i_{thr} \leftarrow \mathsf{argmax}(\mathsf{Diffs})$
6: **return** $\frac{L[i_{thr}+n_{diag}] + L[i_{thr}+n_{diag}+1]}{2}$

---

We choose the birth date threshold as follows: we take the points that are above the persistence threshold and the point with minimal birth date, and sort their birth dates by increasing value in a list $L'$ (indices start at 1). Let $i_{diff}$ be the index such that the birth date difference is maximal and $i_{pers}$ the index of the most persistent point. If $i_{pers} \leq i_{diff}$, then the birth date threshold is set to the birth date of $L'[i_{diff}]$ (this is the typical case where we look for the highest difference). If $i_{diff} = 1$, we consider that all the persistent points are close from one another and are all normal cycles, so we set the threshold to $+\infty$. If $1 < i_{diff} < i_{pers}$, we set the threshold to the birth date of $L[i_{pers}]$, assuming that the most persistent cycle is always normal. See Algorithm 2 for a formal description.

---

**Algorithm 2** BirthDateThreshold (lists start at index 1)

---

**Require:** $\mathsf{Diag}(\mathbb{V})$, $n_{diag}$

1: $L \leftarrow [(b, d) \text{ for } (b, d) \text{ in } \mathsf{Diag}(\mathbb{V})]$
2: $(b_{min}, d_{b_{min}}) \leftarrow \mathsf{min}[b \text{ for } (b, d) \text{ in } L]$ for the lexicographic order.
3: $p \leftarrow \mathsf{PersistenceThreshold}(\mathsf{Diag}(\mathbb{V}), n_{diag})]$
4: $L' \leftarrow [(b_{min}, d_{b_{min}})] + [(b, d) \text{ for } (b, d) \text{ in } L \text{ if } d - b \geq p]$
5: Sort $L'$ using the lexicographic order.
6: $L'_{\text{birth}} \leftarrow [b \text{ for } (b, d) \text{ in } L']$
7: $L'_{\text{pers}} \leftarrow [d - b \text{ for } (b, d) \text{ in } L']$
8: $N' \leftarrow \mathsf{length}(L')$
9: $\mathsf{Diffs}'_{\text{birth}} \leftarrow [L'_{\text{birth}}[i + 1] - L'_{\text{birth}}[i] \text{ for } i = 1 \dots N' - 1]$
10: $\mathsf{Diffs}'_{\text{pers}} \leftarrow [L'_{\text{pers}}[i + 1] - L'_{\text{pers}}[i] \text{ for } i = 1 \dots N' - 1]$
11: $i_{diff} \leftarrow \mathsf{argmax}(\mathsf{Diffs}'_{\text{birth}})$
12: $i_{pers} \leftarrow \mathsf{argmax}(\mathsf{Diffs}'_{\text{pers}})$
13: **if** $i_{diff} == 1$ **then**
14:     **return** $+\infty$
15: **end if**
16: **if** $1 < i_{diff} < i_{pers}$ **then**
17:     **return** $\frac{L'_{\text{birth}}[i_{pers}] + L'_{\text{birth}}[i_{pers} + 1]}{2}$
18: **else**
19:     **return** $\frac{L'_{\text{birth}}[i_{diff}] + L'_{\text{birth}}[i_{diff} + 1]}{2}$
20: **end if**

---

Finally, we keep all points above the persistence threshold and whose birth dates are below the birth date threshold.

If there are no 1-cycles on the diagram, we take an arbitrary "cycle" as normal. In this case, one should look for different parameters $d$ and $\hat{\tau}$ to make the point cloud less dense.

Figure 4 illustrates our algorithm applied to the time series from Figure 1 (with delay embedding from Figure 3). We used $q = 10$, $n_{diag} = 2$ and $n_{points} = 200$. On this example, the birth date threshold is around 2.5, so the two points in the upper left corner are detected as normal (among the 5 most persistent ones).

### 3.2.3   Cycle extraction

The cycle extraction step consists in computing a list of cycles $L_{cycles} = (c_1, \dots, c_{n_{cycles}})$, each one representing the homology class of a cycle identified as a normal at the previous step (a cycle is stored as a list of points).

In practice, we will compute persistent homology with coefficients in $K = \mathbb{Z}/2\mathbb{Z}$. In that case, the following matrix reduction algorithm from Edelsbrunner et al. (2002) can be used.

Let us write our finite filtered simplicial complex as $V = \emptyset = V_0 \subset V_1 \subset \cdots \subset V_N$ such that $V_{i+1} = V_i \cup \sigma_{i+1}$ (where $\sigma_{i+1}$ is a simplex), and let $\partial$ be the $N \times N$ matrix of the boundary operator ($\partial_{i,j} = 1$ if $\sigma_i$ is a face of $\sigma_j$), with columns $C_1 \dots C_N$. Let $\mathsf{low}(C_j)$ be the greatest index $i$ such that $C_j[i] = \partial_{i,j} \neq 0$ (or 0 if the column only has zeros). The matrix reduction algorithm is described in Algorithm 3. The representative cycle corresponding to a point $(l(j), j)$ on the persistence diagram is $\sum_{i=1}^{N} C_j[i]\sigma_i$.

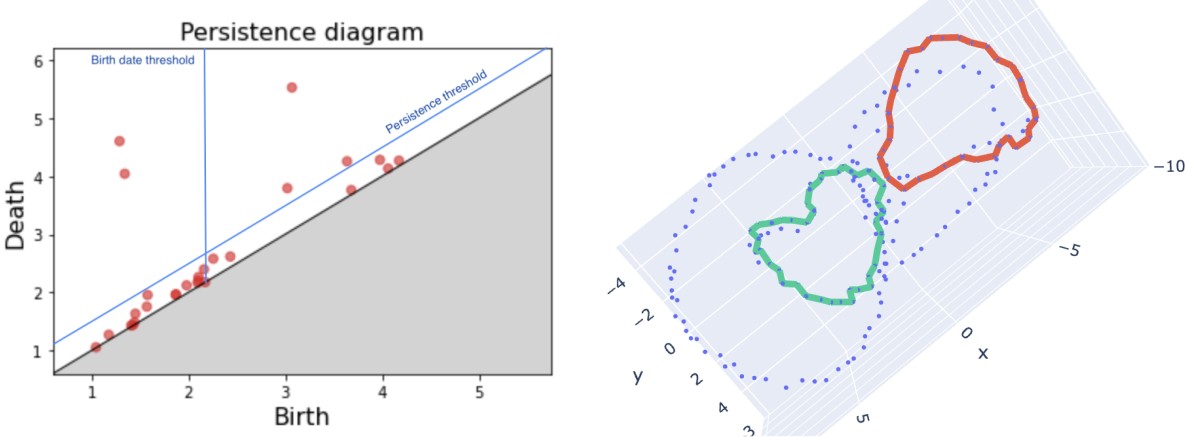

Figure 4: **Left:** persistence diagram of the DTM-filtration of the delay embedding from Figure 3, with persistence and birth date thresholds in blue.. **Right:** subsampling of the delay embedding with 200 points, and cycles detected as normal (in green and red).

---

**Algorithm 3** MatrixReduction

---

**Require:** $\partial = (C_1 \ldots C_N)$
1: **for** $i = 1, \ldots, N$ **do**
2:     **for** $j = N - 1, \ldots, 1$ **do**
3:         **if** $\mathsf{low}(C_i) == \mathsf{low}(C_j) \neq 0$: **then**
4:             $C_j \leftarrow C_i + C_j \ \mathsf{mod}(2)$
5:         **end if**
6:     **end for**
7: **end for**
8: **return** $\partial$

---

The green and red cycles on the point cloud $\tilde{Y}$ from Figure 4 are the two normal cycles extracted with this method, corresponding to the two points evoked above (notice that we found the green and red cycle from Figure 3).

### 3.2.4 Anomaly scores

Once the list $L_{cycles}$ of normal cycles has been computed, an anomaly score is given to each point $x \in \hat{Y}$, which is its distance to $L_{cycles}$: $d(x, L_{cycles})$.

Finally, we get an anomaly score for $\mathbf{y}_i$ by averaging the scores of all points in $\hat{Y}$ of which $\mathbf{y}_i$ is a coordinate: $\mathsf{Score}(\mathbf{y})_i \leftarrow \mathsf{mean}(\{d(\hat{Y}_j, L_{cycles}) \mid \max(0, i - (d-1)\hat{\tau}) \leq j \leq i\})$.

One can choose a threshold to the anomaly score to get a binary answer. Typically, one can chose to keep score only above a certain quantile. In Section 5, we will compare algorithms using the AUC-ROC curve obtained by varying the threshold from 0 to 1 in order not to be biased by an arbitrary choice of threshold. We do not give a specific method to choose the threshold, as it depends on the application (in Section 5, we will compare algorithms using the AUC-ROC curve of each anomaly score not to be biased by an arbitrary choice of threshold). Figure 5 shows the results and ROC curve of our algorithm applied to the signal from Figure 1 (with delay embedding from Figure 3 and normal cycles from Figure 4). The two cycles corresponding to the normal patterns have been extracted so those patterns have an anomaly score close to zero.

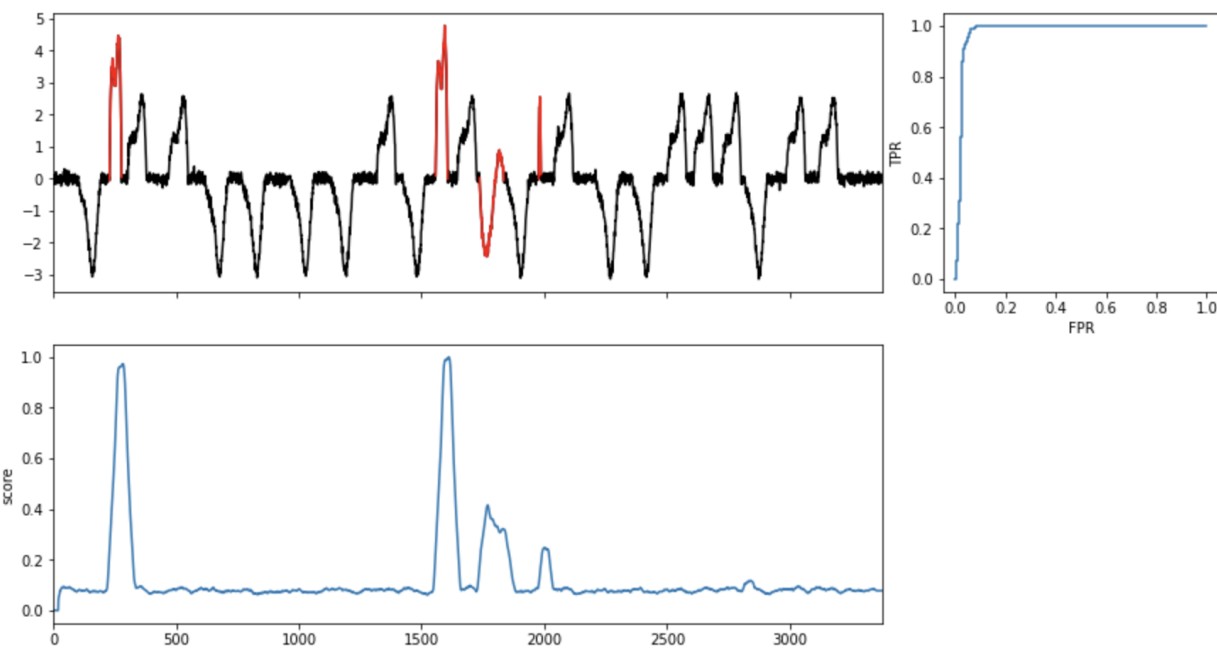

Figure 5: **Top left:** signal from Figure 1 with anomalies in red. **Bottom:** Anomaly score obtained with our algorithm (distance to the cycles from Figure 4, normalized). **Top right:** ROC curve (AUC = 0.98).

## 4 Theoretical study

In this section, we recall stability theorems for DTM filtrations, define a probability measure on delay embeddings of signals and use it (and some additional assumptions) to derive an upper bound on the interleaving distance between the filtration used in our algorithm (with discrete, noisy data with anomalies) and a filtration obtained in an ideal situation with a continuous signal without noise or anomalies.

### 4.1 Overview of existing stability theorems for DTM filtrations

Here, we recall stability results from Anai et al. (2020). In this subsection, $X$ and $Y$ will denote any subsets of $\mathbb{R}^d$. Let $V[X, d_{\mu,m}, p]$ be either the DTM Čech or the DTM Vietoris-Rips filtration, and let $\mathbb{V}[X, d_{\mu,m}, p]$ be the associated persistence module.

**Proposition 2** (Anai et al. (2020), Prop. 3.1). *If $X$ is bounded, $\mathbb{V}[X, d_{\mu,m}, p]$ is q-tame.*

Let $W_2$ denote the *Wasserstein distance with quadratic cost* between $\mu$ and $\nu$ :

$$W_2(\mu, \nu) = \inf_{\pi \in \Pi(\mu,\nu)} \sqrt{\int_{\mathbb{R}^d \times \mathbb{R}^d} ||u - v||^2 d\pi(u, v)}$$

where $\Pi(\mu, \nu)$ is the set of transport plans between $\mu$ and $\nu$ (see Santambrogio (2015) for more details). Let $d_H$ denote the *Hausdorff distance* between sets, defined as

$$d_H(X, Y) = \max\left(\sup_{x \in X} d(x, Y), \sup_{y \in Y} d(X, y)\right).$$

**Proposition 3** (Anai et al. (2020), Prop. 4.3). *Let $\mu_X$ and $\mu_Y$ be two probability measures on $\mathbb{R}^d$ with compact supports $X$ and $Y$. Then*

$$d_i(V[X, d_{\mu_X,m}, p], V[Y, d_{\mu_Y,m}, p]) \leq m^{-\frac{1}{2}} W_2(\mu_X, \mu_Y) + 2^{\frac{1}{p}} d_H(X, Y).$$

The following result is specific to the case $p = 1$ (weaker results are given for $p > 1$ in Anai et al. (2020)).

**Proposition 4** (Anai et al. (2020), Prop. 4.4). *Let $\mu$ be a probability measure on $\mathbb{R}^d$ with compact support* $\mathsf{supp}(\mu)$ *and let* $c(\mu, m) = \sup_{\mathsf{supp}(\mu)} d_{\mu,m}$.

*If* $\mathsf{supp}(\mu) \subset X$ *and* $\mathsf{supp}(\mu) \subset Y$, *then* $V[X, d_{\mu,m}, 1]$ *and* $V[Y, d_{\mu,m}, 1]$ *are $c(\mu, m)$-interleaved and thus*

$$d_i(V[X, d_{\mu,m}, 1], V[Y, d_{\mu,m}, 1]) \leq c(\mu, m).$$

### 4.2 Upper bound on the interleaving distance between the real and ideal filtrations

The idea of our study is to define a filtration of $X_{\mathbf{n}}$, which represents the ideal case of continuous data without any anomaly nor noise, and to show that this filtration is close in terms of interleaving distance to the filtration $V[\hat{Y}, d_{\mu_{\hat{Y}}, m}]$ constructed from the observed data $\hat{Y} = \widehat{X + \mathcal{E}}$ and associated empirical measure. As explained in Section 2, a bound on the interleaving distance between filtration also induces a bound on the interleaving distance between the associated persistence modules and on the bottleneck distance between the associated persistence diagrams, which we use in our algorithm to identify normal cycles. So our algorithm should find cycles that approximate the true normal set $X_{\mathbf{n}}$. We start by making additional assumptions and defining a probability measure $\mu_{X_{\mathbf{n}}}$ on $X_{\mathbf{n}}$, and use the DTM filtration $V[X_{\mathbf{n}}, d_{\mu_{X_{\mathbf{n}}}, m}]$ as our filtration of $X_{\mathbf{n}}$ to state our main result.

#### 4.2.1 Assumption of the theorem and probability measure on curves

In this section, we add some assumption on our model from Section 2 and define a probability measure on $\mathbb{R}^d$ whose support is the curve $X = X_{d,\tau}$ (or $X_{\mathbf{n}}$) that is adapted to the anomaly detection problem, i.e. such that sections corresponding to anomalies have a low probability.

We assume that the delay $\tau$ is small enough so that the curve always returns to 0 between two atoms, and at the beginning and end:

**Assumption 4.** *Let $(\zeta_i)_{1 \leq i \leq M_{\mathbf{n}} + M_{\mathbf{a}}}$ denote all the activations. We assume that:*

1. $\forall i, \forall t \in [0, 1], \quad \zeta_i[t] \neq 0 \Rightarrow \forall j \neq i, \forall s \in [t, t + l_{\zeta_i} + (d-1)\tau], \quad \zeta_j[t] = 0.$

2. $\zeta_i[j] = 0$ *for all $i$ and for all $j \leq (d-1)\tau$ and $j \geq 1 - (d-1)\tau$.*

*Note that a small enough $\tau$ exists because of Assumption 1.*

**Remark 5.** *This assumption ensures that each occurrence of an atom gives (at least) one loop in the delay embedding. It also ensures that each occurrence is complete, which is not mandatory but will make it simpler to compute the probability of each atom.*

Let us now define the probability measure $\mu_X$. Here, $X^{-1}(R)$ denotes the pre-image of $R$ under the function:

$$\begin{cases} [0, 1 - (d-1)\tau] & \to \mathbb{R}^d \\ t & \mapsto X[t] = (\mathbf{x}[t], \mathbf{x}[t + \tau], \dots, \mathbf{x}[t + (d-1)\tau]). \end{cases}$$

**Definition 11.** *Let $\lambda$ denote the Lebesgue measure on $\mathbb{R}$. $\mu_X$ is the Borel measure on $\mathbb{R}^d$ such that for any $d$-rectangle $R = I_1 \times \cdots \times I_d$ where the $I_i$ are open or closed intervals:*

$$\mu_X(R) = \frac{\lambda(X^{-1}(R))}{1 - (d-1)\tau}.$$

It is clear that $\mu_X$ is a probability measure with support $X$.

**Assumption 5.** *let $N_i$ (resp. $A_i$) be the delay embedding of $\mathbf{n}_i * \boldsymbol{\eta}_i$ (resp. $\mathbf{a}_i * \boldsymbol{\alpha}_i$). We assume that the intersection of any pair of sets among the $N_i \setminus \{0\}$ and $A_i \setminus \{0\}$ has null measure.*

In this case, for any $i$, $\mu_X(N_i \setminus \{0\}) = k_{\mathbf{n}_i} l_{\mathbf{n}_i}$ and $\mu_X(A_i \setminus \{0\}) = k_{\mathbf{a}_i} l_{\mathbf{a}_i}$.

**Remark 6.**
- *Assumption 5 makes sense in the context of anomaly detection as it implies that except for a set of null measure, a point $\mathbf{x}_t$ is either normal or abnormal (i.e. normal and abnormal atoms cannot coincide on an interval).*

- *If the atom lengths are equal, then the probability of each atom is proportional to its number of occurrences.*

- *If the signal is constant on an interval, there will be a point with non zero probability in X. It is the case for $0$, which we consider to be part of the normal behavior.*

- *Here, it can already be noticed that the algorithm could fail if an abnormal atom was too long.*

The measure $\mu_{X_{\mathbf{n}}}$ is defined the same way starting from $X_{\mathbf{n}}$ (which is $X$ without the anomalies).

### 4.2.2 Theorem statement

The aim of the section is to prove the following result:

**Theorem 1.** *With the above notations and assumptions, if $\mathbf{x}_{\mathbf{n}}$ is continuously differentiable, and with $p = 1$ for DTM filtrations, we have:*

$$d_i(V[\hat{Y}, d_{\mu_{\hat{Y}}, m}], V[X_{\mathbf{n}}, d_{\mu_{X_{\mathbf{n}}}, m}]) \le (b_n + ||\boldsymbol{\varepsilon}||_\infty)\sqrt{d}\left(m^{-\frac{1}{2}} + 2\right)$$

$$+ m^{-\frac{1}{2}}\sqrt{d}||\mathbf{x}_{\mathbf{a}}||_\infty \sqrt{\sum_{i=1}^{M_{\mathbf{a}}} k_{\mathbf{a}_i} l_{\mathbf{a}_i}}$$

$$+ \sqrt{\frac{d}{3}} \cdot \frac{m||\mathbf{x}'_{\mathbf{n}}||_\infty}{\min_i (k_{\mathbf{n}_i})}$$

*with $b_n \underset{n \to +\infty}{\longrightarrow} 0$.*

Notice that the sampling $\sum_{i=1}^{M_{\mathbf{a}}} k_{\mathbf{a}_i} l_{\mathbf{a}_i}$ is the proportion or the signal that is abnormal, so it can be assumed to be small in the context of anomaly detection. So with a high sampling frequency $n$, a low noise amplitude $||\boldsymbol{\varepsilon}||_\infty$, and if normal atoms occur many times, the bound should be small (after choosing $m$ not too close to zero).

We divide the proof of this theorem in three steps:

- **Sampling step:** Proposition 5 bounds $d_i(V[\hat{X}, d_{\mu_{\hat{X}}, m}, p], V[X, d_{\mu_X, m}, p])$.

- **Noise step:** Proposition 6 bounds $d_i(V[\hat{Y}, d_{\mu_{\hat{Y}}, m}, p], V[\hat{X}, d_{\mu_{\hat{X}}, m}, p])$.

- **Anomaly step:** Proposition 7 bounds $d_i(V[X, d_{\mu_X, m}], V[X_{\mathbf{n}}, d_{\mu_{X_{\mathbf{n}}}, m}])$.

We then conclude using the triangular inequality.

**Remark 7.** *We now make an additional remark regarding the optional subsampling step in our algorithm. Let $\hat{Y}$ be any finite point cloud in $\mathbb{R}^d$. This steps consists in computing the filtration on $\tilde{Y}$, which is a subset of $\hat{Y}$, but we still use the DTM function $d_{\mu_{\hat{Y}},m}$ so Proposition 3 gives:*

$$d_i(V[\hat{Y}, d_{\mu_{\hat{Y}},m}], V[\tilde{Y}, d_{\mu_{\hat{Y}},m}]) \leq 2d_H(\hat{Y}, \tilde{Y}).$$

*In particular, one can take only one point for each occurrence of an atom while still using the information from the number of occurrences through $d_{\mu_{\hat{Y}},m}$ (which would not be the case we used $d_{\mu_{\tilde{Y}},m}$, leading to an important change in the filtration values that would make it hard to discriminate anomalies from normal atoms). This is very useful in practice as computing persistent homology on large datasets can be long.*

**Remark 8.** *Finally, we give a link between this section and Section 3.1. If $m = \frac{q}{n - (d-1)\hat{\tau}}$ with $\mathsf{max}(k_{\mathbf{a}_i}) < q \leq \mathsf{min}(k_{\mathbf{n}_i})$, then for all normal point $y \in Y$, $d_{\mu_{\hat{X}},m}(y) \leq \sqrt{d}(b_n + ||\varepsilon||_\infty)$. So all normal points are dense, and for each abnormal point, at least $q - \mathsf{max}(k_{\mathbf{a}_i})$ neighbors which are considered to compute their DTM do not correspond to the same point from a distinct occurrence. This does not guarantee that abnormal point have a lower density as a single point can appear several times in one occurrence, or the variations can be very slow. So our algorithm can fail to detect an anomaly (even with a small $k_{\mathbf{a}_i}$) if its variations are slow but still forms a large enough 1-cycle to be considered significant. This could especially happen with a large atom length $l_{\mathbf{a}_i}$, which reflects in our model as we approximate a measure $\mu_X$ such that $\mu_X(A_i \setminus 0) = k_{\mathbf{a}_i} l_{\mathbf{a}_i}$.*

### 4.2.3 Sampling step

The next result states that if the signal is well-sampled (i.e. if $n$ is large) then there should be little difference between studying a continuous signal $\mathbf{x}$ or a sampled version $\hat{\mathbf{x}}$. Recall that $\tau = \frac{\hat{\tau}}{n-1}$. For simplicity, we assume that $\frac{1}{\tau} \in \mathbb{N}$, so $\mathsf{Card}(\hat{X}) = \frac{1 - (d-1)\tau}{\tau}$ (points can be counted multiple times).

**Proposition 5** (Stability for sampling). *Let $b_n = \sup\limits_{|t-s| \leq \frac{1}{n-1}} (|\mathbf{x}[t] - \mathbf{x}[s]|)$. With the above notations and assumptions we have:*

$$d_i(V[\hat{X}, d_{\mu_{\hat{X}},m}, p], V[X, d_{\mu_X,m}, p]) \leq b_n \sqrt{d} \left( m^{-\frac{1}{2}} + 2^{\frac{1}{p}} \right).$$

*Proof.* From Proposition 3 we get:

$$d_i(V[\hat{X}, d_{\mu_{\hat{X}},m}, p], V[X, d_{\mu_X,m}, p]) \leq m^{-\frac{1}{2}} W_2(\mu_{\hat{X}}, \mu_X) + 2^{\frac{1}{p}} d_H(\hat{X}, X).$$

It is immediate from the definitions that $d_H(\hat{X}, X) \leq b_n \sqrt{d}$.

Let us now derive an upper bound for $W_2(\mu_{\hat{X}}, \mu_X)$. We refer to Santambrogio (2015); Villani et al. (2009) for and introduction to optimal transport. Let $\lfloor . \rfloor$ denote the floor function, we define $T$ as :

$$T : \begin{cases} X & \to \hat{X} \\ X_t & \mapsto \hat{X}\left[\lfloor (n-1)t \rfloor + 1\right] \end{cases}$$

So $T(X[t]) = \hat{X}[\lfloor (n-1)t \rfloor + 1] = X\left[\frac{\lfloor (n-1)t \rfloor}{n-1}\right]$ with $\left| t - \frac{\lfloor (n-1)t \rfloor}{n-1} \right| < \frac{1}{n-1}$.

We have for any measurable bounded function $f$:

$$\int_{\hat{X}} f(y) d\mu_{\hat{X}}(y) = \sum_{y \in \hat{X}} f(y) \mu_{\hat{X}}(\{y\})$$

$$= \frac{1}{\mathsf{Card}(\hat{X})} \sum_{y \in \hat{X}} f(y)$$

$$= \frac{\tau}{1 - (d-1)\tau} \sum_{y \in \hat{X}} f(y)$$

and

$$\int_X f(T(x))d\mu_X(x) = \sum_{y \in \hat{X}} \int_{\{T^{-1}(y)\}} f(y)d\mu_X(x)$$

$$= \sum_{y \in \hat{X}} f(y)\mu_X(\{T^{-1}(y)\})$$

$$= \sum_{y \in \hat{X}} f(y)\frac{\tau}{1 - (d-1)\tau}.$$

So $\mu_{\hat{X}} = \mu_X \circ T^{-1}$ and $\pi = (\mu_X, \mu_X \circ T^{-1})$ is a (deterministic) transport plan and thus:

$$W_2(\mu_{\hat{X}}, \mu_X) \leq \sqrt{\int_{\mathbb{R}^d \times \mathbb{R}^d} ||u - v||^2 d\pi(u, v)}$$

$$\leq \sqrt{\int_{\mathbb{R}^d} ||u - T(u)||^2 d\mu_X(u)}$$

$$\leq b_n \sqrt{d}.$$

$\square$

### 4.2.4 Noise step

The next result states that the construction is robust to noise.

**Proposition 6** (Stability in the presence of noise)**.** *With the above notations and assumptions we have:*

$$d_i(V[\hat{Y}, d_{\mu_{\hat{Y}}, m}, p], V[\hat{X}, d_{\mu_{\hat{X}}, m}, p]) \leq ||\varepsilon||_\infty \sqrt{d}\left(m^{-\frac{1}{2}} + 2^{\frac{1}{p}}\right).$$

*Proof.* As for the previous proof, we use Proposition 3 to derive a Hausdorff distance term and a Wasserstein distance term. The fact that for all $t$, $||(\widehat{X + \mathcal{E}})_t - \hat{X}_t|| \leq ||\varepsilon||_\infty \sqrt{d}$ immediately implies that $d_H(\hat{X}, \hat{Y}) \leq ||\varepsilon||_\infty \sqrt{d}$. Let $T$ be the function from $\hat{X}$ to $\hat{Y}$ defined by $T(\hat{X}[i]) = \hat{Y}[i]$ so that $\mu_{\hat{Y}} = \mu_{\hat{X}} \circ T^{-1}$ and $\pi = (\mu_{\hat{X}}, \mu_{\hat{X}} \circ T^{-1})$ is a (deterministic) transport plan and thus:

$$W_2(\mu_{\hat{X}}, \mu_{\hat{Y}}) \leq \sqrt{\int_{\mathbb{R}^d \times \mathbb{R}^d} ||u - v||^2 d\pi(u, v)}$$

$$= \frac{1}{\mathsf{Card}(\hat{X})}\sqrt{\sum_{x \in \hat{X}} ||x - T(x)||^2}$$

$$\leq ||\varepsilon||_\infty \sqrt{d}.$$

$\square$

### 4.2.5 Anomaly step

We now fix $p = 1$ and use Proposition 4 to bound the change in interleaving distance between the filtrations obtained respectively with and without anomalies (and the associated DTMs). We note $V[X, d_{\mu_X, m}]$ instead of $V[X, d_{\mu_X, m}, 1]$.

**Proposition 7** (Stability in the presence of anomalies). *With the above notations and assumptions, if $\mathbf{x_n}$ is continuously differentiable:*

$$d_i(V[X_\mathbf{n}, d_{\mu_{X_\mathbf{n}}}, m], V[X, d_{\mu_X}, m]) \leq m^{-\frac{1}{2}}\sqrt{d}\|\mathbf{x_a}\|_\infty \sqrt{\sum_{i=1}^{M_\mathbf{a}} k_{\mathbf{a}_i} l_{\mathbf{a}_i}} + \sqrt{\frac{d}{3}} \cdot \frac{m\|\mathbf{x_n'}\|_\infty}{\min_i(k_{\mathbf{n}_i})}.$$

*Proof.* From Propositions 3 (which still applies in this case where $\mathsf{supp}(\mu_{X_\mathbf{n}}) = X_\mathbf{n} \subset X$ with $X$ compact) and 4 we have:

$$\begin{aligned}
d_i(V[X, d_{\mu_X}, m], V[X_\mathbf{n}, d_{\mu_{X_\mathbf{n}}}, m]) &\leq d_i(V[X, d_{\mu_X}, m], V[X, d_{\mu_{X_\mathbf{n}}}, m]) \\
&\quad + d_i(V[X, d_{\mu_{X_\mathbf{n}}}, m], V[X, d_{\mu_{X_\mathbf{n}}}, m]) \\
&\leq m^{-\frac{1}{2}} W_2(\mu_X, \mu_{X_\mathbf{n}}) + c(\mu_{X_\mathbf{n}}, m).
\end{aligned}$$

Let us first give an upper bound on $W_2(\mu_X, \mu_{X_\mathbf{n}})$. Let $T : X \to X_\mathbf{n}$ be equal to the identity function on $X_\mathbf{n}$ and to the null function on $X \setminus X_\mathbf{n}$. Recall that $X = X_\mathbf{n} \cup X_\mathbf{a}$ and that by assumption $\mu_X(X_\mathbf{n} \cap (X_\mathbf{a} \setminus 0)) = 0$. Also notice that $\mu_X = \mu_{X_\mathbf{n}}$ on $X_\mathbf{n} \setminus 0$.

For any measurable bounded function $f$ we have:

$$\begin{aligned}
\int_X f(T(x)) d\mu_X(x) &= \int_{X_\mathbf{n}} f(T(x)) d\mu_X(x) + \int_{X_\mathbf{a} \setminus 0} f(T(x)) d\mu_X(x) \\
&= \int_{X_\mathbf{n} \setminus 0} f(x) d\mu_X(x) + f(0)\mu_X(0) + \int_{X_\mathbf{a} \setminus 0} f(0) d\mu_X(x) \\
&= \int_{X_\mathbf{n} \setminus 0} f(x) d\mu_{X_\mathbf{n}}(x) + f(0)\mu_X(0) + f(0)\mu_X(X_\mathbf{a} \setminus 0) \\
&= \int_{X_\mathbf{n} \setminus 0} f(x) d\mu_{X_\mathbf{n}}(x) + f(0)\left(1 - \sum_{i=1}^{M_\mathbf{n}} k_{\mathbf{n}_i} l_{\mathbf{n}_i} - \sum_{i=1}^{M_\mathbf{a}} k_{\mathbf{a}_i} l_{\mathbf{a}_i}\right) \\
&\quad + f(0)\left(\sum_{i=1}^{M_\mathbf{a}} k_{\mathbf{a}_i} l_{\mathbf{a}_i}\right) \\
&= \int_{X_\mathbf{n} \setminus 0} f(x) d\mu_{X_\mathbf{n}}(x) + f(0)\mu_{X_\mathbf{n}}(0) \\
&= \int_{X_\mathbf{n}} f(x) d\mu_{X_\mathbf{n}}(x).
\end{aligned}$$

The fourth equality comes from the fact that $\sum_{i=1}^{M_\mathbf{n}} k_{\mathbf{n}_i} l_{\mathbf{n}_i}$ and $\sum_{i=1}^{M_\mathbf{a}} k_{\mathbf{a}_i} l_{\mathbf{a}_i}$ are respectively the amount of time taken by normal and abnormal atoms (the rest of the time is spent at 0), so $\mu_X(0) = 1 - \sum_{i=1}^{M_\mathbf{n}} k_{\mathbf{n}_i} l_{\mathbf{n}_i} - \sum_{i=1}^{M_\mathbf{a}} k_{\mathbf{a}_i} l_{\mathbf{a}_i}$ and $\mu_X(X_\mathbf{a} \setminus 0) = \sum_{i=1}^{M_\mathbf{a}} k_{\mathbf{a}_i} l_{\mathbf{a}_i}$. The fifth equality comes from the fact that when we use $\mu_{X_\mathbf{n}}$, we replace all the abnormal atoms with 0, so $\mu_{X_\mathbf{n}}(0) = 1 - \sum_{i=1}^{M_\mathbf{n}} k_{\mathbf{n}_i} l_{\mathbf{n}_i}$.

The above calculation implies that $\mu_{X_\mathbf{n}} = \mu_X \circ T^{-1}$ and $\pi = (\mu_X, \mu_X \circ T^{-1})$ is a (deterministic) transport plan and thus:

$$W_2(\mu_X, \mu_{X_\mathbf{n}}) \leq \sqrt{\int_{\mathbb{R}^d \times \mathbb{R}^d} ||u - v||^2 d\pi(u, v)}$$

$$\leq \sqrt{\int_{\mathbb{R}^d} ||u - T(u)||^2 d\mu_X(u)}$$

$$\leq \sqrt{\int_{X_\mathbf{a}} ||u||^2 d\mu_X(u)}$$

$$\leq \sqrt{d||\mathbf{x_a}||_\infty^2 \mu_X(X_\mathbf{a})}$$

$$\leq \sqrt{d}||\mathbf{x_a}||_\infty \sqrt{\sum_{i=1}^{M_\mathbf{a}} k_{\mathbf{a}_i} l_{\mathbf{a}_i}}$$

where the fourth inequality comes from the fact that for all $u = (u_1, \ldots, u_d) \in X_\mathbf{a}$ we have: $\forall i,\ u_i \leq ||\mathbf{x_a}||_\infty$.

Let us now compute an upper bound on $c(\mu_{X_\mathbf{n}}, m)$. Let $t, s \in [0, 1 - (d-1)\tau]$. As $\mathbf{x_n}$ is continuously differentiable we have:

$$||X_\mathbf{n}[t] - X_\mathbf{n}[s]||^2 = \sum_{i=0}^{d-1} |\mathbf{x_n}[t + i\tau] - \mathbf{x_n}[s + i\tau]|^2 \leq d|t - s|^2 \cdot ||\mathbf{x_n'}||_\infty^2.$$

Let $u \in ]0, 1[$. We will compute an upper bound on $\delta_{\mu_{X_\mathbf{n}}, u}$ (see Definition 9 for the definition). For all $r \geq 0$ we have:

$$|t - s| \leq \frac{r}{\sqrt{d}||\mathbf{x_n'}||_\infty} \Rightarrow ||X_\mathbf{n}[t] - X_\mathbf{n}[s]|| \leq r$$

so, as there are at least $\min_i(k_{\mathbf{n}_i})$ occurrences of each atom in $\mathbf{x_n}$:

$$\mu_{X_\mathbf{n}}(\bar{B}(X_\mathbf{n}[t], r)) \geq \frac{r \cdot \min_i(k_{\mathbf{n}_i})}{\sqrt{d}||\mathbf{x_n'}||_\infty}$$

and thus:

$$r > \frac{u\sqrt{d}||\mathbf{x_n'}||_\infty}{r \cdot \min_i(k_{\mathbf{n}_i})} \quad \Rightarrow \quad \mu_{X_\mathbf{n}}(\bar{B}(X_\mathbf{n}[t], r)) > u.$$

So we have $\delta_{\mu_{X_\mathbf{n}}, u}(X_\mathbf{n}[t]) \leq \frac{u\sqrt{d}||\mathbf{x_n'}||_\infty}{\min_i(k_{\mathbf{n}_i})}$ and:

$$d_{\mu_{X_\mathbf{n}}, m}(X_\mathbf{n}[t]) = \sqrt{\frac{1}{m} \int_0^m \delta_{\mu_{X_\mathbf{n}}, u}^2(X_\mathbf{n}[t]) du}$$

$$\leq \sqrt{\frac{d}{3}} \cdot \frac{m||\mathbf{x_n'}||_\infty}{\min_i(k_{\mathbf{n}_i})} \ .$$

$\square$

Note that Proposition 4 enabled us to replace $2d_H(X, X_\mathbf{n})$ with $c(\mu_{X_\mathbf{n}}, m)$ in the upper bound (compared to using only Proposition 3). The latter can be equal to $\sqrt{d}||\mathbf{x}||_\infty$ in some cases, whereas the former has a factor $\frac{1}{\min_i(k_{\mathbf{n}_i})}$ that is assumed to be small in our anomaly detection problem.

| Indices | Anomaly type | Several normal atoms | Repeating anomalies |
|---|---|---|---|
| $1-20$ | Punctual | ✗ | ✗ |
| $21-40$ | Sequential | ✗ | ✗ |
| $41-60$ | Punctual | ✓ | ✗ |
| $61-80$ | Sequential | ✓ | ✗ |
| $81-100$ | Sequential | ✓ | ✓ |
| $101-120$ | Both | ✓ | ✓ |

Table 2: Characteristics of each part of the synthetic datasets.

## 5 Experimental setup

This section describes the datasets and anomaly detection methods we will use to test our algorithm in Section 6.

### 5.1 Synthetic dataset

We generated 120 time series following the model described in Section 2. Each atom is generated specifying a random length, choosing a random integer $N$, computing $N$ points on a Gaussian random walk starting at zero, and interpolating them with cubic splines to get an atom of the desired length, which is then multiplied by a random amplitude factor between 0.5 and 2. Atoms length is set to 1 for punctual anomalies and can take values between 150 and 400 for normal atoms and sequential anomalies. $N$ is set to 1 for punctual anomalies and can take values between 5 and 15 for normal atoms and sequential anomalies. For each time series, each normal atom has at most 15 occurrences, and at least 7 more occurrences than the most frequent anomaly: at least 8 when anomalies do not repeat, and 10 when they can have several occurrences (up to 3). Time series have lengths between 1964 and 13234. The dataset can be found at `https://github.com/Alex-B-4/A-persistent-homology-based-algorithm-for-unsupervised-anomaly-detection-in-time-series`.

The dataset is divided in six equal parts, depending on the anomaly type of the time series (punctual or sequential), on the presence or absence of multiple normal atoms and on the possibility for anomalies to have several occurrences (two or three). Table 2 shows the characteristics of each part.

### 5.2 Real-world datasets and methods from TSB-UAD

Our real-world datasets are the 18 public datasets provided by the TSB-UAD benchmark suite (Paparrizos et al., 2022), for a total of 1980 univariate time series with labeled anomalies. Table 3 shows the names, sizes (i.e. number of time series), average lengths and data types of 18 datasets.

For each time series, we compute the area under the ROC curve (AUC-ROC) obtained by looking at all the possible thresholds on the anomaly score. The same method is applied to 13 anomaly detection algorithms (Liu et al., 2008; Breunig et al., 2000; Goldstein & Dengel, 2012; Yeh et al., 2018; Boniol et al., 2021; Aggarwal & Aggarwal, 2017; Sakurada & Yairi, 2014; Malhotra et al., 2015; Li et al., 2007; Munir et al., 2018; Schölkopf et al., 1999; Lu et al., 2022) in Paparrizos et al. (2022). Table 4 provides a succinct description of the methods. Using the AUC-ROC, the evaluation does not depend on the choice of a threshold for each algorithm. The datasets and the implementations of the 13 methods can be found at `https://github.com/TheDatumOrg/TSB-UAD/tree/main`.

## 6 Results

In this section, we start by showing the results our the benchmark on real-world datasets. Then, we use the synthetic dataset to study the influence of our five parameters and of noise on the performance and computation time of our algorithm.

| Name | Size | Average lengths | Data type |
|---|---|---|---|
| Dodgers | 1 | 50400.0 | Freeway traffic |
| ECG | 53 | 230351.9 | Electrocardiogram |
| IOPS | 58 | 102119.2 | Performance indicators for machines |
| MGAB | 10 | 100000.0 | Differential equations (used for biological models) |
| NAB | 58 | 6301.7 | Server metrics and internet traffic |
| NASA-MSL | 27 | 2730.7 | Spacecraft telemetry |
| NASA-SMAP | 54 | 8066.0 | Spacecraft telemetry |
| Sensorscope | 23 | 27038.4 | Environmental data |
| YAHOO | 367 | 1561.2 | Metrics on Yahoo services |
| KDD21 | 250 | 77415.06 | Composite dataset |
| Daphnet | 45 | 21760.0 | Acceleration sensors on Parkison's disease patients |
| GHL | 126 | 200001.0 | Metrics on Gasoil reservoirs |
| Genesis | 6 | 16220.0 | Industrial time series |
| MITDB | 32 | 650000.0 | Electrocardiogram |
| OPP | 465 | 31616.9 | Motion sensors on humans |
| Occupancy | 10 | 5725.8 | Environmental data |
| SMD | 281 | 25562.3 | Server metrics |
| SVDB | 115 | 230400.0 | Electrocardiogram |

Table 3: Characteristics of the real-world datasets from TSB-UAD.

| Method | Unsupervised | Delay embedding | Description |
|---|---|---|---|
| TDA | ✓ | ✓ | Our method: TDA to find normal 1-cycles |
| IForest1 Liu et al. (2008) | ✓ | x | Random forest on time series to find outliers |
| IForest Liu et al. (2008) | ✓ | ✓ | Random forest on delay embedding to find outliers |
| LOF Breunig et al. (2000) | ✓ | ✓ | Outlier score based on local density |
| HBOS Goldstein & Dengel (2012) | ✓ | ✓ | Outlier score based on histograms |
| MP Yeh et al. (2018) | ✓ | x | Distance between closest subsequence as anomaly score |
| DAMP (Lu et al., 2022) | ✓ | x | A version of the Matrix Profile designed to better handle multiple anomalies |
| NORMA Boniol et al. (2021) | ✓ | x | Clustering to find normal subsequences |
| PCA Aggarwal & Aggarwal (2017) | ✓ | ✓ | PCA to define a normal hyperplane |
| AE Sakurada & Yairi (2014) | x | x | Reconstruction error as anomaly score, based on deep learning |
| LSTM Malhotra et al. (2015) | x | x | Prediction error as anomaly score, based on deep learning |
| POLY Li et al. (2007) | ✓ | x | Prediction error as anomaly score, based on polynomial interpolation |
| CNN Munir et al. (2018) | x | x | Prediction error as anomaly score, based on deep learning |
| OCSVM Schölkopf et al. (1999) | x | ✓ | SVM to find normal points on delay embedding |

Table 4: Overview of the anomaly detection methods used in our benchmark. The "Delay embedding" column indicates if the methods uses a delay embedding or not.

## 6.1 Benchmark on real-world data

Here, we show the results of our algorithm on the TSB-UAD benchmark described in Section 5.2. The parameters of our algorithm are chosen the same way for all time series. We estimate a period $L$ for the time series using the first maximum of the autocorrelation function (we used the find_length function from TSB-UAD, which was used for all other methods using a delay embedding). We empirically chose $\hat{\tau} = 6$ (in practice, if $\hat{\tau}$ is too small, the embedding will stay close to the line spanned by $(1, \ldots, 1)$ and it will be harder to detect cycles). In Perea & Harer (2015) Perea and Harer show that in the case of trigonometric functions, $d\hat{\tau}$ should be a multiple of the period to maximize persistence in 1D homology. With this in mind, we empirically set $d = \max(40, \min(120, \lfloor \frac{L}{3} \rfloor))$. We chose $q = \lfloor \frac{n}{L} \rfloor$ following the intuition from Section 3.1, as this value would approximate the number of normal occurrences in the case where there is one normal atom. Finally, we use $n_{diag} = 2$ and $n_{points} = 400$.

The algorithm was implemented in Python. The DTM filtration was computed with the GUHDI library (Maria et al., 2014). The cycle extraction algorithm was implemented with dionysus[1](more specif-

---

[1]https://pypi.org/project/dionysus/

|  | TDA | IForest | IForest1 | LOF | MP | PCA | NORMA | HBOS | POLY | OCSVM | AE | CNN | LSTM | DAMP |
|---|---|---|---|---|---|---|---|---|---|---|---|---|---|---|
| Dodgers | **0.79** | **0.79** | 0.64 | 0.54 | 0.52 | 0.77 | **0.79** | 0.3 | 0.69 | 0.64 | 0.73 | 0.68 | 0.39 | 0.62 |
| ECG | 0.88 | 0.75 | 0.61 | 0.56 | 0.58 | 0.71 | **0.95** | 0.68 | 0.70 | 0.64 | 0.73 | 0.52 | 0.54 | 0.72 |
| IOPS | **0.82** | 0.54 | 0.78 | 0.50 | 0.72 | 0.74 | 0.76 | 0.64 | 0.68 | 0.71 | 0.63 | 0.61 | 0.61 | 0.50 |
| MGAB | 0.58 | 0.57 | 0.58 | **0.96** | 0.91 | 0.54 | 0.55 | 0.54 | 0.51 | 0.52 | 0.71 | 0.58 | 0.56 | 0.55 |
| NAB | **0.76** | 0.45 | 0.56 | 0.48 | 0.49 | 0.69 | 0.58 | 0.68 | 0.75 | 0.61 | 0.54 | 0.52 | 0.50 | 0.54 |
| NASA-MSL | 0.64 | 0.57 | 0.69 | 0.52 | 0.52 | 0.75 | 0.55 | 0.77 | **0.81** | 0.64 | 0.70 | 0.57 | 0.57 | 0.67 |
| NASA-SMAP | **0.83** | 0.72 | 0.68 | 0.68 | 0.62 | 0.74 | 0.80 | 0.77 | 0.80 | 0.65 | 0.77 | 0.68 | 0.64 | 0.77 |
| SensorScope | 0.52 | 0.56 | 0.56 | 0.55 | 0.50 | 0.54 | 0.59 | 0.56 | **0.62** | 0.51 | 0.52 | 0.52 | 0.53 | 0.56 |
| YAHOO | 0.64 | 0.62 | 0.81 | 0.86 | 0.86 | 0.57 | 0.92 | 0.57 | 0.76 | 0.50 | 0.79 | **0.96** | 0.94 | 0.78 |
| KDD21 | 0.75 | 0.65 | 0.57 | 0.78 | **0.90** | 0.58 | 0.88 | 0.60 | 0.58 | 0.60 | 0.79 | 0.74 | 0.66 | 0.76 |
| Daphnet | 0.70 | 0.74 | 0.68 | **0.78** | 0.44 | 0.69 | 0.46 | 0.69 | 0.77 | 0.45 | 0.44 | 0.47 | 0.44 | 0.50 |
| GHL | 0.86 | **0.94** | **0.94** | 0.54 | 0.42 | 0.91 | 0.64 | 0.92 | 0.76 | 0.45 | 0.63 | 0.47 | 0.47 | 0.53 |
| Genesis | 0.84 | 0.78 | 0.66 | 0.68 | 0.35 | 0.85 | 0.6 | 0.59 | **0.87** | 0.70 | 0.72 | 0.73 | 0.53 | 0.71 |
| MITDB | 0.71 | 0.70 | 0.61 | 0.61 | 0.69 | 0.67 | **0.86** | 0.70 | 0.68 | 0.65 | 0.80 | 0.58 | 0.51 | 0.66 |
| OPP | 0.48 | 0.49 | 0.52 | 0.45 | **0.82** | 0.52 | 0.65 | 0.54 | 0.28 | 0.38 | 0.70 | 0.47 | 0.57 | 0.65 |
| Occupancy | 0.53 | 0.86 | 0.78 | 0.53 | 0.32 | 0.78 | 0.53 | **0.89** | 0.80 | 0.66 | 0.69 | 0.79 | 0.71 | 0.53 |
| SMD | 0.77 | 0.85 | 0.73 | 0.69 | 0.51 | 0.80 | 0.61 | 0.77 | **0.87** | 0.61 | 0.63 | 0.61 | 0.58 | 0.57 |
| SVDB | 0.77 | 0.72 | 0.58 | 0.59 | 0.74 | 0.68 | **0.92** | 0.71 | 0.67 | 0.68 | 0.79 | 0.58 | 0.55 | 0.70 |

Table 5: Average AUC-ROC on each dataset. Results for methods other than TDA come from Paparrizos et al. (2022).

ically, with a modified version of cyclonysus[2]). The code can be found at `https://github.com/Alex-B-4/A-persistent-homology-based-algorithm-for-unsupervised-anomaly-detection-in-time-series`. Results for the other methods come from Paparrizos et al. (2022) except for DAMP, for which we used the implementation from TSB-UAD.

Table 5 shows the average AUC-ROC obtained on each dataset with our method (TDA) with the above parameters, and the results of the TSB-UAD benchmark (Paparrizos et al., 2022). Figure 6 shows the critical diagrams comparing the average rank of each method using the Friedman test followed by the Wilcoxon or Nemenyi test with $\alpha = 0.05$, as described in Demšar (2006).

These results show that our method is competitive with the state-of-the-art in anomaly detection on 18 standard datasets. It has the best score on 4 of them, the best average rank (though the difference with the best methods is not significant as shown on Figure 6), and it is in the top 5 on 13 datasets. We found three reasons to explain what can make our algorithm perform well or not. First, our algorithm was designed for the problem described in Section 2, which makes it appropriate for structured data with repetitive patterns but datasets can differ from our model in several ways: the absence of a clear normal behavior, a trend, or a lack of continuity (for example binary time series or time series with very fast variations) or too much noise can lead to a delay embedding with no relevant normal cycles to detect. Anomalies can also have a nature that our algorithm cannot detect, such as a longer pause between two atoms. The second reason is the choice of parameters. Even though our heuristics to choose the parameters seem relevant (see Sections 6.2 and 6.3), it is still not optimal to have the same rule to choose parameters for datasets with different types of data, lengths, sampling rates... The delay embedding parameters $d$ and $\hat{\tau}$ should not be too low or else the curve will not have enough space to form persistent cycles, but if they are too high points of the delay embedding will represent different atoms and the structure will be higher to detect. Moreover, if $d\hat{\tau}$ is significantly larger than the length of an anomaly, there will be false positive because the algorithm will detect too many indices as abnormal. We believe that $q$ and $n_{diag}$ have less influence on performance (see Sections 6.2 and 6.3). The third reason is the quality of the subsampling $\tilde{Y}$. If the time series is short, or if it has many repetitive patterns, then 400 points can be enough to represent the whole point cloud, but many time series have more than 100000 points so if they have numerous and/or long atoms then it is not enough and the distance between points will be too low for cycles to persist (this is studied in more details in Section 6.4.

---

[2]`https://github.com/sauln/cyclonysus`

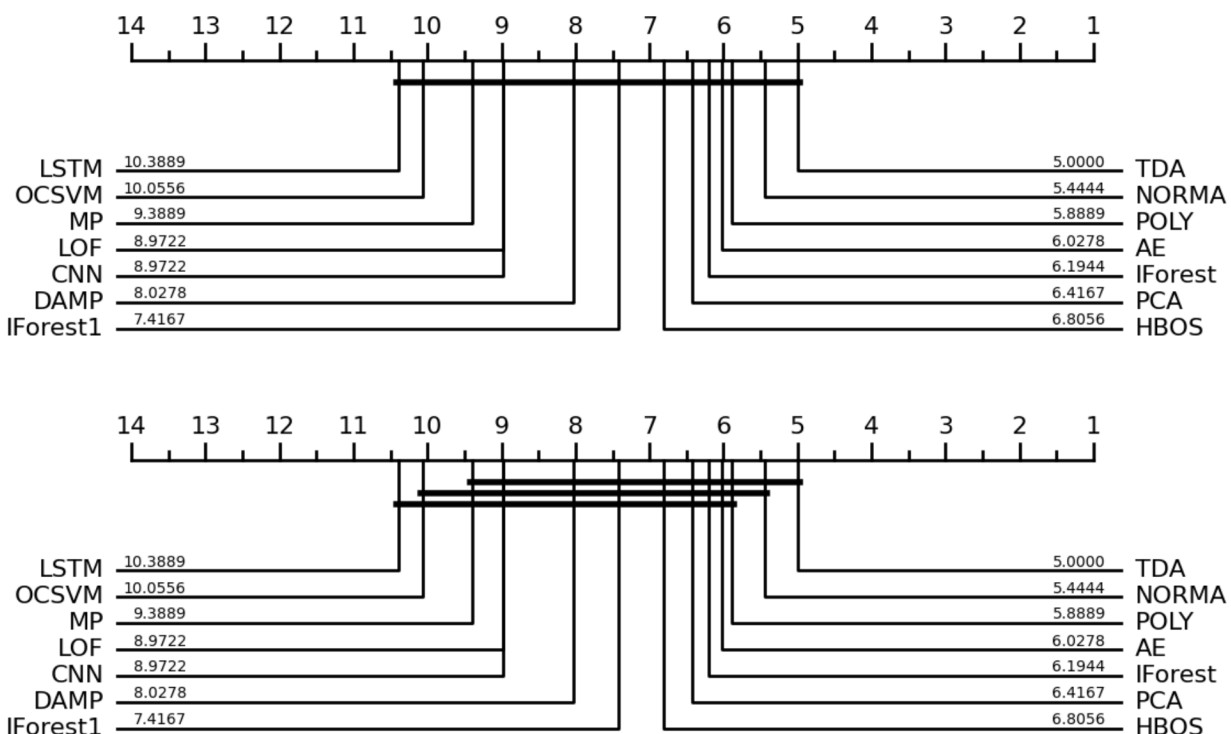

Figure 6: Critical diagrams for $\alpha = 0.05$. **Top:** Friedman test + Wilcoxon test. **Bottom:** Friedman test + Nemenyi test.

## 6.2    Influence of the embedding dimension and delay

Here, we study the influence of the delay embedding parameters $d$ and $\hat{\tau}$ by looking at the performance of our algorithm on our synthetic dataset with different pairs of parameters $(d, \hat{\tau})$, while the other parameters are fixed. Figure 7 is a heat map that shows the average AUC-ROC on the whole dataset when $d$ varies between 3 and 70 and $\hat{\tau}$ varies between 1 and 11. We fixed $n_{points} = 100$ and $n_{diag} = 2$. We set $q = \lfloor \frac{n}{L} \rfloor$, where $L$ is found using the autocorrelation function, as in Section 5.

On Figure 7, there is one zone where the AUC-ROC is above 0.81 and the AUC-ROC decreases as parameters go away from this zone (below or above), going to the $0.78 - 0.81$ zone then to the $0.75 - 0.78$ zone... until reaching 0.6 at the top right and bottom left corners. The shape of the zones $AUC \geq 0.81$, $AUC \geq 0.78$, and $AUC \geq 0.75$ and of their frontiers suggest that the best scores are obtained when the product $d\hat{\tau}$ is constant. This product represent the time window represented by each point of the delay embedding. The window should be large enough to capture complex patterns but the window should not cover several atoms. In our dataset, atom lengths can have random values between 50 and 400. Figure 7 suggests that $d\hat{\tau}$ should be approximately between 40 and 80. This empirically confirms the intuition from Section 5 and Perea & Harer (2015) that the product $d\hat{\tau}$ should be of the order of the atom lengths. Moreover, an AUC-ROC above 0.84 is never reached with $d$ above 30 or below 10, nor with $\hat{\tau}$ above 8 or below 2. This indicates that each parameter should not take extreme values. Indeed, for a given value of the product $d\hat{\tau}$, if $\hat{\tau}$ is too small then the delay embedding will be concentrated around the line $x_1 = x_2 = \cdots = x_d$, and if it is too high then points on the embedding will not represent the local variations that constitue the shape of the atoms.

## 6.3    Influence of the number of neighbors and points on the diagram

Here, we study the parameters $q$, which has an influence on the filtration values and thus on the persistence diagram, and $n_{diag}$, which influences the way we read the persistence diagram. We look at the performance

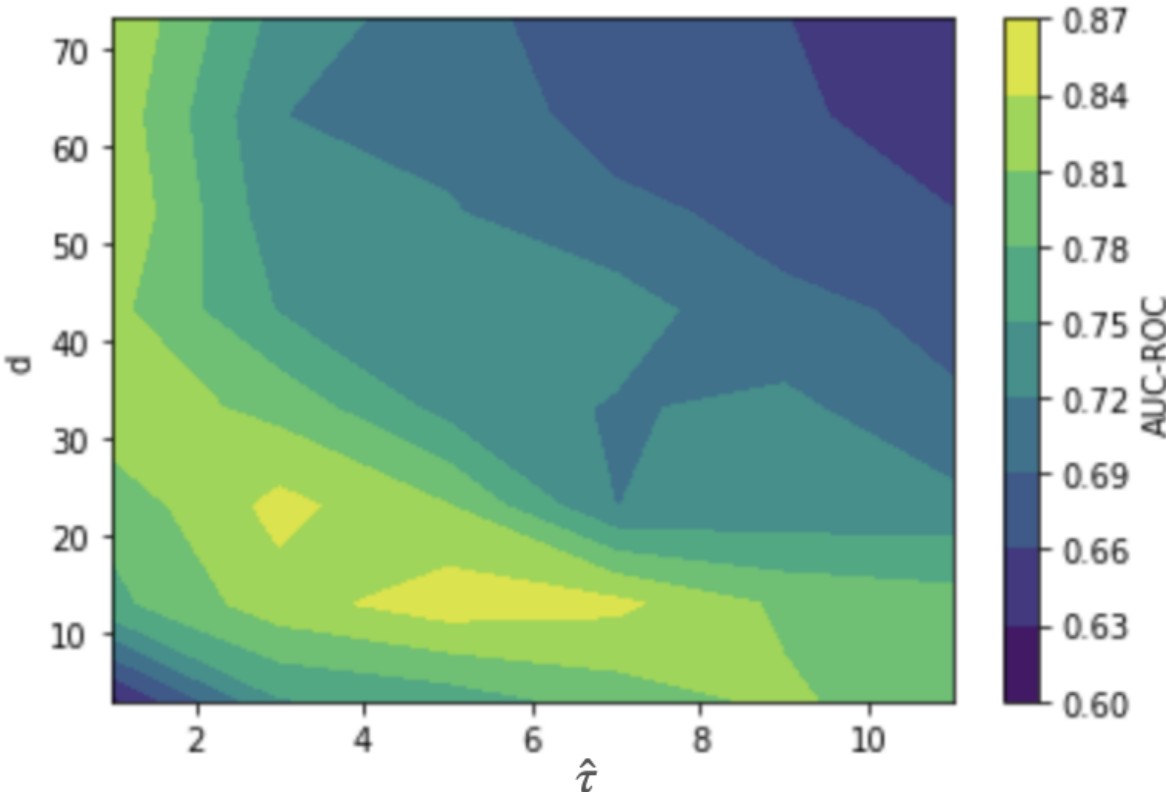

Figure 7: Heat map representing the average AUC-ROC on the whole dataset with different values of $d$ and $\hat{\tau}$.

of our algorithm on our synthetic dataset with different pairs of parameters $(q, n_{diag})$, while the other parameters are fixed. Figure 8 shows the average AUC-ROC on the whole dataset when $q$ varies between 1 and 140 and $n_{diag}$ varies between 2 and 16 (as explained in 3.2.2, we always set $n_{diag} \geq 2$). We fixed $n_{points} = 100$, $d = 15$ and $\hat{\tau} = 5$.

On Figure 8, we can see zones of constant AUC-ROC separated by almost horizontal lines, except for $2 \leq n_{diag} \leq 6$ where the zone $AUC = 0.92$ gets thinner as $n_{diag}$ decreases. Moreover, the AUC-ROC significantly decreases when $q$ gets lower than 20. This suggests that the algorithm is robust to changes of $q$ and $n_{diag}$ except for very low values of $q$ (the proposed heuristic makes $q$ proportional to $n$ to avoid this situation), and that setting $n_{diag}$ a little higher than 2 can make it easier to reach optimal performance, but considering more cycles can make it harder to find the right birth date threshold.

## 6.4 Computation time and influence of the subsampling parameter

A nice property of our method is the possibility of computing persistent homology on a subset of the point cloud while keeping density information about the whole point cloud in the filtration. This makes the complexity of the algorithm go from $O(n^3)$ (Edelsbrunner et al., 2002) to $O(n_{points}^3 + n^2)$, as persistent homology is computed on a point cloud of size $n_{points}$, but the distance matrix of the full points cloud must be computed. Figure 9 shows the computation time when $n_{points}$ varies from 50 to 600 (on average on the 20 first time series of the dataset), on a normal and logarithmic scale (to empirically confirm the theoretical complexity), along with the AUC-ROC (to study the compromise between computation time and performance). We set $d = 15$, $\hat{\tau} = 15$, $q = 50$ and $n_{diag} = 2$.

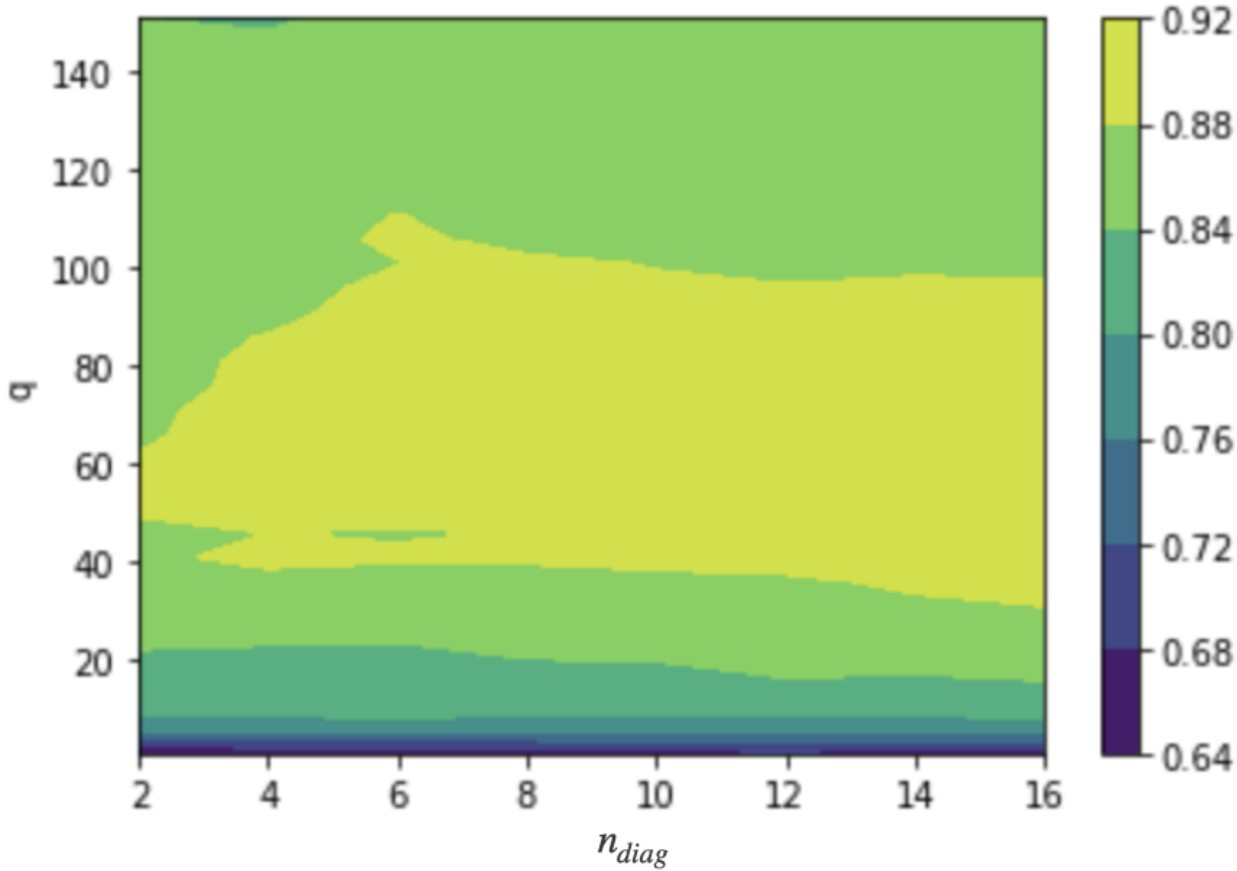

Figure 8: Heat map representing the average AUC-ROC on the whole dataset with different values of $q$ and $n_{diag}$.

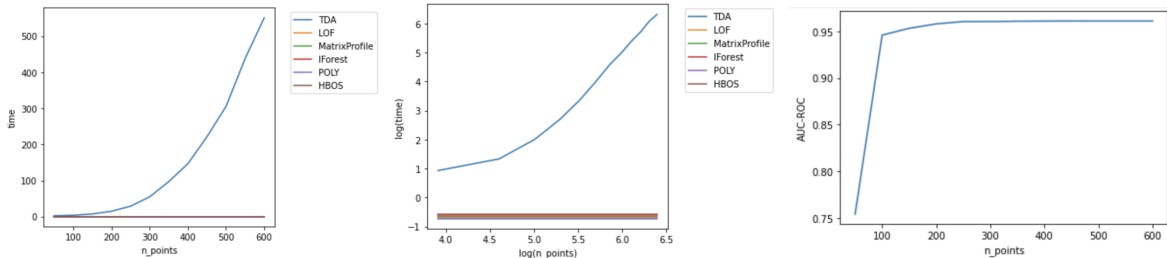

Figure 9: **Left:** computation time (in seconds) as a function of $n_{points}$. **Middle:** computation time as a function of $n_{points}$, logarithmic scale. **Right:** AUC-ROC as a function of $n_{points}$. All quantities are averaged over 20 time series.

On Figure 9, it appears that the AUC-ROC stops increasing after $n_{points} = 200$, at a value above 0.95. Computing persistent homology without subsampling (i.e. on thousands of points) would take hours for each time series, but these examples show that performance can be almost optimal for values of $n_{points}$ significantly smaller than $n$ (here, of the order of 10 times smaller, which makes computation about 1000 times faster). This study, and the fact that our algorithm performed well with $n_{points} = 400$ on time series with lengths above 100000 in our benchmark from Section 6.1, indicate that the algorithm has a certain scalability as with $n_{points}$ can increase slower than $n$.

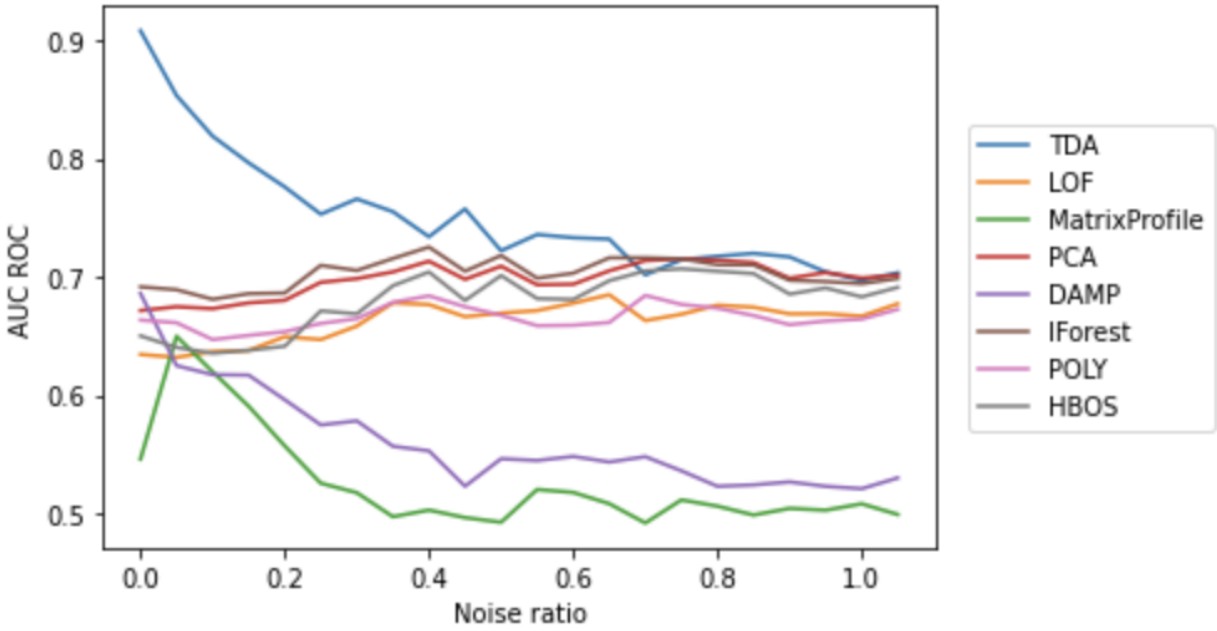

Figure 10: Average AUC-ROC of each algorithm on the whole synthetic dataset, when the noise ratio of the centered Gaussian noise varies between 0 and 1.

## 6.5 Robustness to noise and benchmark on synthetic data

Here, we study the robustness of our algorithm to noise, and compare it to the unsupervised methods from the benchmark of Section 5 whose code is publicly available, and DAMP (Lu et al., 2022). For each signal $\mathbf{y} = \mathbf{x} + \varepsilon$, we define the *noise ratio* as $\frac{\mathrm{Var}(\varepsilon)}{\mathrm{Var}(\mathbf{x})}$ (the inverse of the signal-to-noise ratio). Figure 10 shows the average AUC-ROC of each algorithm on the whole synthetic dataset, where centered Gaussian noise is added to each time series, with a noise ratio between 0 and 1. Note that the real benchmark is in Section 6.1, we only make comparisons here to gain insights our algorithm. It should be noted that since the synthetic data are generated according to the exact model used by our method, the comparison is biased.

Figure 10 shows that our algorithm clearly outperforms all the other unsupervised methods up to a noise ratio of 30%, and stays in the top performers for higher noise ratios (it seems that the AUC-ROC of about 0.7 corresponds to a points where only easy anomalies are detected, which would explain the fact that it can be reached by all the algorithms, even with a lot of noise). When looking into more details at our study on synthetic data, it appears that our algorithm usually performs better than the others when there are multiple normal atoms or repeating anomalies. However, LOF can outperform it in the case of punctual anomalies, especially when the normal behavior is complex (several atoms or a lot of noise). Also note that the significant decline in performance of the Matrix Profile and DAMP in the presence of noise can be explained by the fact that they use a z-normalized Euclidean distance, which can amplify the noise when looking at sequences between atoms so those sequences are detected as anomalies and can even hide other anomalies.

Our method gets an AUC-ROC above 0.8 for noise ratios up to approximately 20%. Too much noise can fill the loops so the main cycles are less persistent and thus more difficult to identify on the persistence diagram. All the other methods except the Matrix Profile and DAMP are very robust to noise. However, it would seem more relevant to study those methods on data for which they perform very well to see how much noise makes them fail.

## 7 Conclusion

This article describes an unsupervised anomaly detection algorithm based on 1D persistent homology. We proposed a model of time series in which the anomaly detection problem is well defined, which enabled us to create our method and to derive mathematical properties of the method. The algorithm empirically proved to be competitive to state-of-the-art anomaly detection methods on different real-world datasets. The subsampling step induces a time/performance compromise that enabled us to significantly decrease computation time. Future research could improve compromise by using faster ways of extracting cycles or finding sparse versions of the filtrations. Another research perspective would be to adapt the algorithm to more general models of time series, for example to deal with the presence of a linear trend, or with multivariate time series.

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
