# OpenReview forum: "A persistent homology-based algorithm for unsupervised anomaly detection in time series"
_TMLR — Accepted by TMLR_

### Review · Reviewer_Nnzu · 2024-08-31

**Summary Of Contributions:**

The authors introduce a new TSAD algorithm based on delay imbedding

**Audience:**

Yes

**Broader Impact Concerns:**

no issues

**Claims And Evidence:**

No

**Requested Changes:**

I would like to see experiments on meaningful datasets.

Such datasets exist, but they take a little more effort to find (the kdd21 is a good start).

 Most of the dataset you consider are either tiny, trivial or mislabeled.   Think about it, if 20% of the data is mislabeled, you cannot say "we are 5% better than algorithm A!"

**Strengths And Weaknesses:**

For my comments below I am relying on Wu and Keoghs
A) Current Time Series Anomaly Detection Benchmarks are Flawed and are Creating the Illusion of Progress. IEEE Trans. Knowl. Data Eng. 35(3): 2421-2429 (2023)
B)  Their frequent Linkined postings on datasets, which they archive here. https://www.dropbox.com/scl/fi/cwduv5idkwx9ci328nfpy/Problems-with-Time-Series-Anomaly-Detection.pdf?rlkey=d9mnqw4tuayyjsplu0u1t7ugg&st=bkj3i9wn&dl=0
However, with my students, I have largely confirmed their claims.
--
My main problem with your paperis that I don’t believe your experiments show the utility of your ideas, because..

1)	Look at SVDB. 56.5% of the data is anomalous!!!  This is a simple classification problem, it is not a TSAD problem.
2)	Look at YAHOO. The synthetic subset can be solved with one line of code, and the real subset has many false positives and false negatives in the ground thruth
3)	MITDB you can solve with fast parameter-free (and over 20 years old) time series discords, you don’t need a complex algorithm.
4)	Dodgers has huge problems with its ground truth, it has many false negatives
5)	SensorScope has dozens of false negatives in the ground truth
6)	Look at Genesis.  There are long runs of 0,0,0,0,0,0,0,0,0,0,0,0,0,0,…   and some of these are anomalies and some are not.  It is really hard to believe that the ground truth is meaningful here, and I would be VERY suspicious of any algorithms that did well on this (without access to out of band data)
7)	What about IOPS?  Wu and Keogh show you can solve it with this single line of code plot(diff(IOPS(1:end,1))>10);       Again, you don’t need a complex algorithm
8)	What about GHL?    Again Wu shows you can solve it with one line of code       plot(zscore((LevcorrTempfaultseed199vars23(:,6)))>1.1)
I think the problem is that you lean uncritically on Paparrizos et al. But he assembled these datasets with no real thought or introspection (other than trying to cripple the Matrix Profile /repeated anomalies)

•	The best dataset you consider is the kdd21, where you do worse than the MP.
•	You appear to do well on Dodgers, but look at slide 45 of [B], the ground truth here is better than random, but is clearly has LOTs of false positives.
•	You appear to do well on MSL, but look at https://arxiv.org/pdf/2009.13807 fig 9. There are at least two false negatives in this single subset.
•	You appear to do well in IOPS, but we can solve that with one line of code, how excited should we be?
•	Etc etc

In summary, your experiments do not offer forceful evidence that your algorithm is competitive with much simpler ideas like the MP or iForest. Maybe it is, but we cannot tell from these experiments. If you look at how kdd21 was created, it does seem that this is a very high quality datasets, and we should weight its evidence more, which I am inclined to do.




The discussion of “repeated anomalies.” is problematic.   Paparrizos and others like to pretend it is a problem for the Matrix Profile, but I think DAMP finally kills that.  However, “repeated anomalies.” Can greatly inflate an algorithms performance by overcounting success.
Suppose you have one second video (30 fps) of Brad Pitt smiling, and you run an algorithms on it to classify the face. Can you say “we got 30 out of 30 correct!!”?  Clearly no, these are not independent, if you get any one correct, you will surely get all correct.   Repeated anomalies are the same. You can take credit for the first one you find, but after that you are just taking credit for the same thing.

Along those lines, you should compare to left-Matrix Profile (DAMP) not classic Matrix Profile if you are going to include the datasets (like ecg) that Paparrizos included to try to make his method better look good relative to the MP

our fundamental idea is “Points in those cycles are considered as normal, and the algorithm can then assign an anomaly score to any point which is its distance to the normal set.”  It is similar to time series discords “Subsequences (cycles) close to other subsequences are considered as normal, and the algorithm can then assign an anomaly score to any subsequence which is its distance to nearest other Subsequence.”, it would be good to compare/contrast.


In table 5 you mention “ECG” dataset, that is a little vauge as there are many such datasets with that name.
You show that for MGAB, the Matrix profile is goodish, but not perfect (table 5), if you look at slide 65 of [B] that is surprising, as the Matrix Profile is perfect on MGAB

---

> ### Author Response · Authors · 2024-11-20
> **Answer to reviewer Nnzu**
>
> We thank the reviewers for their comments. We are aware of the current debate within the data mining research community about benchmarks and datasets for anomaly detection. Some researchers have pointed out that some of the datasets in this benchmark are too easy, while others argue that the aim of an anomaly detection algorithm is to be able to handle all situations, including the easiest ones.
>
> We don't want to enter into this debate: our paper is devoted to geometry and topological data analysis and, in our opinion, our performance on data sets represents only a small part of our contribution. The application of TDA tools to time series analysis is in itself a very great challenge, and we believe that the ideas developed in our paper are of great interest to researchers in signal processing, geometry and applied mathematics (which is the scientific community targeted by the paper). We were able to derive a model and prove some theoretical results, which we believe is the main scientific content that should be judged.
>
> That said, in addition to the synthetic experiments that carefully discuss all the parameter tuning issues and give an overview of the “good” and “bad” use cases of our method, we wanted to show some results on real data. The benchmark we used has the advantage of grouping different data sets into a single package and making our results reproducible. We included it to present a more objective comparison with the state of the art than our experiments on synthetic data and to provide a “rough” estimate of our method's performance on real use cases. This experimental part is not the most important for us, as this is not an article on data mining, and at no point in the article do we claim that our algorithm is the best and should be the only method to use in the future. We only state that it is competitive, which is sufficient for the TMLR criteria of technical soundness and reasonable claims.
>
> In the reviewer's comments, some of the 18 datasets are considered too easy or flawed. Yet we don't have access to any other data (and it seems that there are no other easily available datasets for anomaly detection at present - as the reviewer rightly points out). So, if we understand correctly, what the reviewer is asking us to do is to delete most of the datasets and include only the KKD21 dataset, which, according to the reviewer, is the only “valid” dataset in this benchmark. While we are willing to mention in our article that certain performances should be treated with caution due to known issues, removing results from our benchmark is a very important change that we cannot make unless all three reviewers agree to do so. Indeed, some reviewers might have other views on the matter and might consider that removing 17 data sets constitutes fraud. We'd prefer to keep all the results (with a possible disclaimer) and let readers judge based on those they deem relevant. But of course, if all the reviewers want us to remove all 17 datasets, we will.
>
> Following one of the reviewer’s remarks, we will add the DAMP algorithm to our benchmark in the final version of the paper. The experiments are still running and we will provide the whole results in the next few days.

---

> > ### Author Response · Authors · 2024-11-22
> > **DAMP added to the benchmark.**
> >
> > As announced in our previous comment, we have added the DAMP algorithm to our benchmark and we have changed the article accordingly. This addition does not change our claims or our conclusions.
> >
> > Please find below the average AUC ROC for DAMP and our method on each dataset:
> > | Datasets    | DAMP |  TDA |
> > |-------------|:----:|:----:|
> > | Dodgers     | 0.62 | 0.79 |
> > | ECG         | 0.72 | 0.88 |
> > | IOPS        | 0.50 | 0.82 |
> > | MGAB        | 0.55 | 0.58 |
> > | NAB         | 0.54 | 0.76 |
> > | NASA-MSL    | 0.66 | 0.64 |
> > | NASA-SMAP   | 0.77 | 0.83 |
> > | SensorScope | 0.56 | 0.52 |
> > | YAHOO       | 0.78 | 0.64 |
> > | KDD21       | 0.76 | 0.76 |
> > | Daphnet     | 0.50 | 0.71 |
> > | GHL         | 0.53 | 0.86 |
> > | Genesis     | 0.71 | 0.84 |
> > | MITDB       | 0.66 | 0.71 |
> > | OPP         | 0.65 | 0.48 |
> > | Occupancy   | 0.53 | 0.53 |
> > | SMD         | 0.57 | 0.77 |
> > | SVDB        | 0.70 | 0.77 |

---

> ### Comment · Reviewer_Nnzu · 2024-12-05
> **I greatly appreciate the great efforts to include DAMP and the response. And my opinion of the work has increased.**
>
> I greatly appreciate the great efforts to include DAMP and the response. And my opinion of the work has increased.
> However, I still have reservations, as I note below. Do with them what you will.
>
>
> I don’t feel comfortable asking you to remove all 17 datasets.
> However, look at Dodgers www.youtube.com/watch?v=iRN5oVNvZwk& It really is the case that the labels are basically random, there are so many false positives and many false negatives in the ground truth that the labels are basically random.
>
>
> Look at IOPS. The anomalies are almost all huge spikes that you can detect with simple statistical processing control (SPC) algorithms. From Keogh’s tutorial handouts, this solves it perfectly.
>
> plot(diff(KPI05f10d3a239c3bef9bdca2feeb0037aa(1:end,1))>10);
>
> and so on
>
> --
>
> “ while others argue that the aim of an anomaly detection algorithm is to be able to handle all situations, including the easiest ones.”
>
> This is NOT a fair argument.
>
> For example, in “Dodgers” people take credit for finding the anomalies of negative values (for missing data). But that is a known (ahead of time) formatting convention, not something anyone needs to learn.
> In some of your datasets, the anomalies is a several order of magnitude change in value, they can be filtered out with a line of code, a SPC.
> My point is that we should not take credit for finding simple trivial anomalies, that could be found with faster, similar decades old methods
>
> https://www.linkedin.com/posts/philipp-leser-52b93ba6_anomaly-detection-systems-activity-7267255980827021312--8UJ?utm_source=share&utm_medium=member_desktop
>
> https://www.linkedin.com/posts/stefansuwelack_is-ml-based-anomaly-detection-a-hoax-dear-activity-7264223317367111680-5iZ0/?utm_source=share&utm_medium=member_desktop

---

### Review · Reviewer_R1QX · 2024-11-01

**Summary Of Contributions:**

The paper describes an unsupervised anomaly detection algorithm based on 1D persistent homology. To be detailed, the authors first proposed a time series model that formally define the anomaly detection problems. Given this model, a new anomaly detection algorithm is proposed based on persistent homology, which relies on delay embeddings and the Vietories-Rips filtration. Theoretical analysis for the proposed method is established. Finally, the behavior of the proposed method is studied. Overall, the paper is well-written and the studied topic is interesting.

**Audience:**

Yes

**Claims And Evidence:**

Yes

**Requested Changes:**

1. See weakness (2).

2. Assumptions 1-3 in Section 2 need some discussion.

3. What does "uniform noisy subsampling" mean in Assumption 2?

4. The paper assumes that all the atoms start and end at zero. What if some atoms don't start and end at zero?

5. At the beginning of Section 2.2.1, $|| . ||$ should be $|| \cdot ||$.

**Strengths And Weaknesses:**

Strengths

A new algorithm based on 1D persistent homology associate with theoretical analysis is proposed for analyzing anomaly detection problems.

Weakness

1. The problem setting considered in this paper is limited. e.g., it focuses on unsupervised anomaly detection in univariate time series, while it would be more interesting to consider more general cases, i.e., multivariate time series. In addition, the model established in the paper is not suitable for many types of data and there are many restrictive assumptions.

2. The experiments do not fully demonstrate the effectiveness and efficiency of the proposed algorithm. It seems that many existing methods outperform the proposed method. Specifically, first, in Table 5, NORMA and POLY seem to have better prediction behavior than TDA. In particular, for the ECG dataset, NORMA has an AUC-ROC score of 0.95, while TDA has an AUC-ROC score of 0.88. This is a very unreasonable result because according to Section 2, ECG is a dataset that matches the model established in this paper, but TDA's results on this dataset are not as good as NORMA. The effectiveness of the proposed algorithm needs to be verified. Secondly, the paper points out that "One advantage of our method is that persistent homology can be calculated on a subset of the point cloud while retaining the density information of the entire point cloud in filtering. This reduces the complexity of the algorithm from O(n^3) to O(n_points^3 + n^2)...". Section 6.4 provides some discussion and illustrations in this direction, but this is not enough to prove the computational efficiency of the proposed method. It would be better to compare the computation time with state-of-the-art methods. Also, there are a lot of hyperparameters to choose from and I'm not sure it's easy to choose.

---

> ### Author Response · Authors · 2024-11-20
> **Answer to reviewer Reviewer R1QX**
>
> We thank the reviewer for their review. Our answers to the requested changes are below.
>
>         1. See weakness (2).
>
> Performances of the algorithm.
> Our algorithm comes out first on four datasets, and so do NORMA and POLY. Our algorithm also comes out first on average with no significant differences with the other algorithms except for LSTM and OCSVM. Consequently, we believe that our method is competitive with other methods in the literature. Another aspect is that, unlike most anomaly detection methods (including deep learning approaches), our anomaly detector is linked to a model that allows intuitive interpretation by practitioners. In essence, our algorithm detects typical patterns and defines anomalies as patterns that deviate from these typical patterns. Building our algorithm on such an intuitive and easy-to-understand principle could be a very interesting aspect for use in real-life scenarios (arguably more important than purely score-related aspects).
>
> Performances on the ECG dataset.
> Thanks to the reviewer’s comment about the ECG dataset, we investigated further and found that the mistakes on the ECG dataset (where our method performs very well with an AUC of 0.88 but less than NORMA, as pointed out by the reviewer) can be explained by the fact that there are variations of amplitude in the heartbeats (corresponding to high noise in our model) and very fast variations. These two conditions can make it hard for the algorithm to detect relevant loops because holes in the delay embedding are filled or points are too far away to approximate a continuous curve.
> Following this remark, we added the following explanation to the end of section 6.1:
> « the absence of a clear normal behavior, a trend, or a lack of continuity (for example, binary time series or time series with very fast variations) or too much noise can lead to a delay embedding with no relevant normal cycles to detect .»
>
> Computational time.
> Computation time is indeed an important issue, which led us to design a method to compute a filtration on a subsampled point cloud while keeping information from the whole data. This strategy allowed us to significantly decrease computational time, and we discuss this question in detail in Section 6.4, where we study the influence of the subsampling strategy on the performances.
> Following the reviewer’s remark, we added to Section 6.4 an experiment showing the computation time of state-of-the-art methods to compare it to ours (for increasing values of n_points). Our experiment shows that the algorithm can perform even with a small n_points. Indeed, when applied to time series of thousands of points from our synthetic datasets, we obtained AUC-ROCs around 0.95 as soon as n_points was over 150 with only a slight improvement for higher values of n_points, which indicates that in this case, the result obtained with 150 points was close to the result we would obtain without any subsampling. This would still be true for very long time series if they are made of many patterns of reasonable length. Indeed, in that case, each normal loop would have many occurrences but only one would be needed by our algorithm.
> If n becomes so large that computing the full distance matrix becomes prohibitive, an idea would be to use a truncated time series or sliding windows of reasonable length to detect normal cycles and to only compute the anomaly scores on the rest of the data. This would be efficient if the truncated time series had enough occurrences of normal patterns.

---

> ### Author Response · Authors · 2024-11-20
> **Answer to reviewer R1QX (2)**
>
> 2. Assumptions 1-3 in Section 2 need some discussion.
>     4. The paper assumes that all the atoms start and end at zero. What if some atoms don't start and end at zero?
>
> We answer points 2 and 4 at the same time.
> The assumptions of our model are indeed important both for the design of the algorithm and the theory. The first one implies that all the atoms start and end at zero. The delay embedding of a continuous function made of patterns that start and end at the same point (we chose zero without loss of generality) is a set of closed curves, which is what our TDA algorithm detects.
> The second one is a common assumption in modelling. The fact that the sampling is uniform is useful for the theoretical part as the coefficient b_n used in the bound is related to this assumption, but is not crucial for the design of the algorithm.
> The third one corresponds to the specificity of the anomaly detection problem (compared, for example, to the pattern detection problem), which is that anomalies are rare compared to normal behavior.
>
>
> Following the reviewer’s remark, we added the following discussion of the three assumptions in Remark 2 of the revised manuscript:
>
> Assumption 1 implies that all the atoms start and end at zero. This will be crucial for our method as we will transform the signal into a curve on which we will look for loops. The assumption ensures that each pattern corresponds to at least one loop.
>
> Assumption 2 corresponds to the situation where a quantity assumed to vary continuously is measured at regular intervals, with noise corresponding to errors or perturbations.} Note that the only assumption we will make on the noise is that $||\boldsymbol{\varepsilon}||_\infty$ is small compared to the variations of the signal. Thus, it could represent small variations at each occurrence of an atom or noise due to the data acquisition process. Moreover, introducing $\boldsymbol{\varepsilon}$ also makes it possible to study the problem without the disjoint supports assumption as long as the overlap between supports causes a small enough change in the infinity norm of $\mathbf{x}$.
>
> Assumption 3 corresponds to the specificity of the anomaly detection problem (compared, for example, to the pattern detection problem).
>
>     3. What does "uniform noisy subsampling" mean in Assumption 2?
>
> It means that y is a uniform subsampling (they are defined on page 4) of a noisy version of the signal x. We rephrased this in the revised manuscript.
>
>     5.       At the beginning of Section 2.2.1….
>
> We have corrected this typo.

---

### Review · Reviewer_PodD · 2024-11-05

**Summary Of Contributions:**

Authors study time series anomaly detection using Topological Data Analysis (TDA). They provide an algorithm that can detect different types of anomalies and is based on loop detection in persistent homology. Their theorems, algorithms and techniques are applicable generally and require only mild assumptions. Furthermore, they run experiments on many synthetic and real data sets to show the effectiveness of their approach.

**Audience:**

Yes

**Claims And Evidence:**

Yes

**Requested Changes:**

1) Please clarify the main technical differences from (Anai et al., 2020). Many definitions and theorems of this paper rely on (Anai et al.,2020).

2) Authors mentioned that (Anai et al., 2020) are able to perform changepoint detection for timeseries. Can the authors describe why one cannot use their method as a subroutine, perform a sliding window routine across the time series and call their subroutine to perhaps identify minimal intervals where the changepoint occurs. The proposed method is not efficient, however it would great if the authors could better highlight the subtle differences in changepoint detection and anomaly detection.

3)  Related to point 2), it would be great if the authors can describe how their approach (either algorithmically or theoretically) different and better suited for anomaly detection.

4) Can this algorithm be implemented in cases of sensor networks where the nodes have limited computational resources and the anomalies have to be flagged immediately?

**Strengths And Weaknesses:**

Strengths:

1) The time series model is general and the assumptions are reasonable.

2) They are able to use an anomaly detection method based on TDA which can deal with various types of anomalies in the dataset, from burst to spikes, repeated and non-repeated, as long as they are not too large a fraction of the total data points the time series array.

3) They have theoretical results that guarantee that their algorithms identify approximately the "right" normal parts of a time series. For this they on techniques from Optimal Transport (under mild assumptions).

4) The experimental results on many real data sets beat various other approaches (that include DNN based ones).

Weaknesses:

1) The paper is likely to be hard to follow for an average ML researcher. While the authors have tried to provide intuition, some of the mathematical background related to persistent homology could go in the appendix. I would believe highlighting the algorithm is perhaps more useful to the majority of the ML community interested in anomaly detection.

2) A lot of the results depend on prior work (Anai et al., 2020) and in places it is not clear the differences in contributions of this paper vs (Anai et al., 2020). I have added a few points regarding this in the "Requested Changes" section below.


References:
Anai, Hirokazu, et al. "DTM-based filtrations." Topological Data Analysis: The Abel Symposium 2018. Springer International Publishing, 2020.

---

> ### Author Response · Authors · 2024-11-20
> **Answer to reviewer PodD**
>
> We thank the reviewer for their review. Our answers to the requested changes are below.
>
>          1. Please clarify the main technical differences from (Anai et al., 2020). Many definitions and theorems of this paper rely on (Anai et al.,2020).
>
> In their original paper, Anai et al., 2020 introduced the DTM filtration, which is a filtration that integrates information about the density of each data point (more precisely, the density is measured by the mean squared distance to a certain number of nearest neighbors of each point). The methods and theoretical results they obtain are recalled in our paper in Sections 2.2 and 4.1. Based on your comments, we have added a sentence to specify that these sections constitute background content.
>
> In our proposed method, we indeed rely on the DTM to discriminate normal cycles from abnormal ones based on their birth date, which reflects their density (as normal cycles should be denser). However :
> We modified the DTM filtration in order to adapt it to our task: we compute a subsampled point cloud and assign to each point a filtration value corresponding to its value in the DTM filtration of the whole point cloud.
> We have used the DTM filtration as a tool to identify normal cycles but then proposed to address the anomaly detection task, which was not done in Anai’s original paper, and derived a model and new theoretical results. Therefore, most ideas presented in our paper are completely novel.
>
> Following this remark, we added the following sentence to the revised manuscript:
> « Note that sections 2.2, 2.3, and 4.1 present already existing objects and results. Every other section describes new research. »
>
>
>         2. Authors mentioned that (Anai et al., 2020) are able to perform changepoint detection for time series. Can the authors describe why one cannot use their method as a subroutine, perform a sliding window routine across the time series and call their subroutine to perhaps identify minimal intervals where the changepoint occurs. The proposed method is not efficient, however it would great if the authors could better highlight the subtle differences in changepoint detection and anomaly detection.
>
>         3. Related to point 2), it would be great if the authors can describe how their approach (either algorithmically or theoretically) different and better suited for anomaly detection.
>
> We answer points 2 and 3 at the same time.
> As explained in the paper, anomaly detection consists in detecting rare events in a time series. Change point detection consists in detecting changes from one regime to another. If we consider a time series from our model, a changepoint detection algorithm could detect the start of a new pattern, but it would not make the difference between a normal and an abnormal one (every occurrence of a pattern would constitute a change). Moreover, the three change point detection methods we evoke in the introduction (note that methods from Anai et al. and Fernández et al. are not explicitly described in the papers but presented as examples) are based on computing the persistence diagram corresponding to a sequence of growing subwindows of a time series, before comparing consecutive diagrams to detect changes. When there are multiple patterns, changes would be detected at the first occurrence of each pattern but they would not be detected at each next occurrence. In the case of a repeated anomaly, this would lead to false negatives, and in the case of multiple normal patterns, this would lead to false positives.
> Moreover, this sliding window-based change detection method requires to compute many persistence diagrams, which can lead to a high computation time. Our method relies on only one persistence diagram representing the whole time series and integrating density information about points of the delay embedding. This density reflects the number of occurrences of each pattern which is how we differentiate normal ones from abnormal ones.
>
> Following this remark, we added the following explanation to the introduction:
> « Indeed, they consist in computing the persistence diagram corresponding to a sequence of growing subwindows of a time series and comparing consecutive diagrams to detect changes. Thus, when there are multiple patterns, changes would be detected at the first occurrence of each pattern but they would often not correspond to an anomaly. In the case of a repeated anomaly, a change would only be detected at its first occurrence, and the following ones would be missed. Moreover, this method requires to compute many persistence diagrams, which can lead to a high computation time. »

---

> > ### Author Response · Authors · 2024-11-20
> > **Answer to reviewer PodD (2)**
> >
> > 4. Can this algorithm be implemented in cases of sensor networks where the nodes have limited computational resources and the anomalies have to be flagged immediately?
> >
> > The complexity of our algorithm is O(n_points^3 + n^2), where n is the length of the time series and n_points is the number of points of the subsampled point cloud. Section 6.4 of our paper presents a study of the computation time of our algorithm as a function of n_points (with n fixed). As seen in Section 6.4, the sub-sampling strategy introduced in our paper enables us to reduce computation time considerably, and in our experiments we were able to process time series of up to more than 600 000 points.
> >
> > Our algorithm, as well as all the methods presented in our benchmark, are offline algorithms that cannot detect anomalies on the fly and require knowledge of all the data to calculate anomaly scores. However, most methods (including ours) could be adapted to the online setting by using sliding windows (e.g. buffers). This strategy would be possible with our method, provided the buffer size is large enough to hold multiple occurrences of the “normal patterns”. It's also worth noting that the use of smaller sliding windows (compared to long time series) would considerably reduce computation time and enable “light” implementation. So, although this is future work, there is no scientific lock on extending our framework to an online, low-computing environment.

---

### Author Response · Authors · 2024-11-20
**General answer to reviewers**

First of all, we would like to thank the reviewers for their comments, which helped us improve the article and better explain our contributions and claims.

We answered each reviewer individually and made a revised version of the manuscript.
We have made the following changes to the paper:

- In the introduction, we clarified the differences between changepoint and anomaly detection methods and precisely identified which objects from Anai et al. (2020) we used and which ones are our contributions.
- In section 2.1, we discussed the assumptions of our model.
- In section 6.1, we will add the DAMP algorithm to our benchmark.
- In section 6.4, we added a computation time comparison of our method to several state-of-the-art methods and explained in more detail some sources of error for our method.

---

### Decision · Action_Editor_DKAQ · 2024-12-23

**Recommendation:** Accept as is

**Comment:**

Two reviewers recommend "leaning accept" while the third recommends "leaning reject".   One reviewer is concerned with the difference between the proposed approach and Anal et al. (2020), the reviewer didn't have further comments after the author's response.   Another reviewer was concerned with the datasets used, while the authors didn't add more datasets, they add one more existing method DAMP, which the reviewer seems satisfied and recommends "leaning accept."

While the proposed method is competitive, not better than, with existing methods, their topological analysis approach is interesting, which could provide a different perspective to the community.

**Audience:**

Part of the TMLR audience would find the article interesting, particularly the anomaly detection community.

**Claims And Evidence:**

For univariate time series anomaly detection, the authors propose using Topological Data Analysis (TDA).  Particularly, they use delay embeddings and persistent cycles from the 1-dimensional persistent homology based on Vietoris-Rips filtration.  The four main steps are compute the DTM Rips filtration of a subset of the delay embedding, identify the normal 1-cycles, extract them, and compute the anomaly scores.

The proposed TDA approach is compared with 13 existing methods over 18 datasets from the TSB-UAD benchmark.   The results indicate that the TDA approach is competitive with the existing methods.